# Synovial sarcoma reprograms transcription by GBAF activation of polycomb targets and loss of CBAF enhancers

Jinxiu Li[1,2,8], Li Li[1,2,8], Kyllie Smith-Fry[1,2], Muhammad Zaki Fadlullah [2], Lara Carroll[1,2], Linda Morrison[1,2], Xinyi Ge[1,2], Mary L. Nelson [1,2], Lesley A. Hill [3], Yixuan Guo [2], George Davenport[4], Xiaoyang Zhang [2], Torsten O. Nielsen [4], Martin Hirst[5,6], T. Michael Underhill [3], Bradley R. Cairns [2,7] ✉ & Kevin B. Jones [1,2] ✉

Synovial sarcoma is a cancer driven by a fusion oncoprotein, SS18::SSX, that links SS18, a subunit of BAF-family chromatin remodeling complexes, to the carboxy terminus of SSX, which avidly binds nucleosomes with the histone post-translational modification H2AK119ub. Here, we show in mice that SS18::SSX expression redistributes non-canonical GBAF complexes broadly to promoters and distal enhancers marked by H2AK119ub, which causes developmental loci to lose H3K27me3 and become transcriptionally active. Canonical BAF containing SS18::SSX abandons its typical binding sites, is largely absent from H2AK119ub-marked sites, and instead distributes narrowly to transcription start sites with PBAF. Disruption of Arid1a or Arid1b (both CBAF-specific) retains synovial sarcoma character, while Smarcb1 (PBAF- and CBAF-specific) or Pbrm1 (PBAF-specific) disruption does not, although all accelerate SS18::SSX-driven tumorigenesis in mice. Thus, the synovial sarcomagenesis mechanism involves SS18::SSX reprogramming transcription positively through GBAF redistribution to activate polycomb-targeted developmental genes, and negatively by loss of normal CBAF localization and function.

Synovial sarcoma (SyS) is a dangerous malignancy of the soft-tissues, primarily affecting adolescents and young adults[1]. Initially characterized by the expression of epithelial cell markers, SyS arises in mesenchymal tissue compartments[2]. SyS consistently associates with a chromosomal translocation between chromosomes 18 and X, which produces one of three specific fusion genes: *SS18::SSX1*, *SS18::SSX2*, or *SS18::SSX4*[3,4]. In mice, expression of one of these fusion oncogenes (*SS18::SSX1* and *SS18::SSX2* have been tested) in mesenchymal progenitors independently initiates tumorigenesis,

strongly recapitulating human SyS, histopathologically and molecularly[5–7].

Defining the targets and mechanisms of transcriptional reprogramming by chromatin-associated fusion oncoproteins presents many challenges[8–11]. Transcriptional signatures have been identified as shared among large cohorts of human SySs[12–15] and even shared between mouse and human sarcomas initiated by SS18::SSX expression[5,6,16], but the ideal control cell type/condition for comparison has not been available. Differential expression between SyS cell

[1]Department of Orthopaedics, Spencer Fox Eccles School of Medicine, University of Utah, Salt Lake City, UT, USA. [2]Department of Oncological Sciences, Spencer Fox Eccles School of Medicine, Huntsman Cancer Institute, University of Utah, Salt Lake City, UT, USA. [3]Department of Cellular and Physiological Sciences, University of British Columbia, Vancouver, BC, Canada. [4]Department of Pathology, University of British Columbia, Vancouver, BC, Canada. [5]Canada's Michael Smith Genome Sciences Centre, BC Cancer, Vancouver, BC, Canada. [6]Department of Microbiology and Immunology, Michael Smith Laboratories, University of British Columbia, Vancouver, BC, Canada. [7]Howard Hughes Medical Institute, University of Utah, Salt Lake City, UT, USA. [8]These authors contributed equally: Jinxiu Li, Li Li. ✉e-mail: brad.cairns@hci.utah.edu; kevin.jones@hci.utah.edu

lines with depleted or intact fusion expression introduces the potential artifact that removing fusion expression does not necessarily revert a cell to its pre-transformed state[17,18]. These fusion oncoproteins may powerfully reprogram transcription in ways similar to Yamanaka factors, which prompt a transition to an epigenetically induced cellular state that, for at least part of the transcription program, may no longer require the expression of the reprogramming factors. Interrogating the distribution of SS18::SSX across chromatin genome-wide in cell lines has also been difficult. It has generally been performed using CRISPR/Cas9-inserted tags into the fusion oncogene itself, which may or may not introduce artifactual adjustments to its distribution and function[19,20]. From these varied sources of uncertainty, competing models for transcriptional impact of SS18::SSX have been proposed, ranging from transcriptional activation solely through direct oncoprotein targeting (with indirect transcriptional repression), or a mixture of direct and indirect targets activated and repressed by the fusion.

The oncoprotein that drives SyS consists of the SSX carboxy terminal tail that recognizes/binds nucleosomes with ubiquitylated histone H2A (H2AK119ub)[15,21–23], fused to the SS18 component of both canonical (CBAF, also cBAF) and non-canonical (GBAF, GLTSCR1-containing, also ncBAF) BAF-family chromatin remodeling complexes (Fig. 1a)[20,24–27]. In non-SyS contexts, H2AK119ub, a polycomb repressive complex (PRC1)-placed post-translational modification associates with the PRC2-placed transcription-silencing trimethylation of lysine 27 on histone H3 (H3K27me3, Fig. 1b)[15,21,22,28]. Developmental genes that are poised for transcription, but not actively transcribed, have promoters marked by both transcription-repressing H3K27me3 and transcription-enabling trimethylation of lysine 4 on histone H3 (H3K4me3, Fig. 1c)— termed "bivalent" promoters[29–31]. Conversely, the promoters of actively transcribed genes bear H3K4me3 accompanied by acetylated lysine 27 of histone H3 (H3K27ac, Fig. 1d)[32]. Active enhancers (distal from the promoters they enhance) are typically marked by mono-methylation of lysine 4 of histone H3 (H3K4me1) as well as H3K27ac (Fig. 1e)[33]. BAF complexes remodel (slide or eject) the nucleosomes marked by these varied post-translational modifications, with the polybromo subtype (PBAF) focused primarily at transcription start sites (TSSs) in promoters and CBAF and GBAF distributed predominantly to distal enhancer loci.[25] PRC2-placed H3K27me3 is anti-correlated with the binding of BAF family remodelers[34], whereas H3K4me3 and H3K27ac marks are generally correlated. Thus, the SS18::SSX oncoprotein fuses a component of CBAF and GBAF (SS18) to a protein domain (SSX) that binds it directly to nucleosomes marked for silencing, creating a chromatin conflict.

We explore the mechanism of this conflict, revealing how the fusion impacts CBAF and GBAF differently to result in the inappropriate activation of developmental genes that define and promote SyS. The study of synovial sarcomagenesis in the mouse has provided a tractable and faithful platform for experimentation, testing elements of the biology observed in human SyS[7,14,16,35–37]. Although epigenomics assessments of whole tumors add computational complexity from the infiltrating immune, endothelial, and supporting stromal cells, such non-tumor cell populations comprise a small fraction of the overall cells in SyS in humans and mice[38], permitting robust interrogation of biology using these methods applied to whole tumors.

Here, we begin from a foundation set by comparing single cell transcriptomes from a discrete cell of origin to cellular states that punctuate the range of states across SyS development that are consistent between mice and humans and explored in detail in companion articles (Fig. 1f-j)[29,31]. Through this approach, we generate a catalog of genes associated with sarcomagenesis, facilitating the assignment of discrete roles to mapped SS18::SSX positions across the genome. Having previously established that the observed altered prevalence of BAF subtypes in SyS cells is achieved by the incorporation of SS18::SSX into both GBAF and CBAF[7], we here leverage these insights to help determine the basic role of BAF complexes in SyS transcriptional reprogramming.

## Results

### SS18::SSX distributes to proximal and distal regulatory elements bearing ubiquitylated histone 2 A

In order to evaluate SS18::SSX distribution across chromatin, genome-wide, we performed immunoprecipitation followed by DNA sequencing of chromatin samples (ChIP-seq) from mouse SyS tumors expressing human *SS18::SSX2* (herein denoted as *hSS2*), using an antibody specific for the fusion oncoprotein[39]. SyS tumors were generated by expressing the fusion in mouse hindlimb tissues, initiated by TATCre protein injection at 8 days of life (Fig. 1j). Tumors included a range of histomorphologies across the spectrum of SyS (monophasic, biphasic, and poorly differentiated). ChIP-seq was also employed to assess SyS genome profiles of specific chromatin histone marks, and chromatin conformation by HiChIP was evaluated using an antibody against H3K27ac to identify enhancer loops for the purpose of assigning distal regulatory elements to specific genes (Supplementary Fig. 1a-d).

In our associated paper, we defined three categories of genes determined by comparison of single cell RNA sequencing (scRNA-seq) of mesenchymal stromal cells that give rise to SyS upon expression of the fusion (MSC, cells of origin) and fully transformed SyS cells (Fig. 1h)[31]. These gene categories comprise: 1) genes that transition from no significant expression in MSCs to expression in SyS (sarcomagenesis activated transcription, SAT genes), 2) genes that are expressed in both the MSCs and the transformed SyS cells (maintained active transcription, MAT genes), and 3) genes that transition from active expression in MSCs to negligible expression in transformed SyS cells (sarcomagenesis silenced transcription, SST genes).

In order to identify direct targets of SS18::SSX among the genes that increase expression with sarcomagenesis, we evaluated each of the promoters in the SAT genes for the fusion oncoprotein ChIP-seq enrichment surrounding the transcription start site (TSS ± 2kB) and the distal anchors of HiChIP loops that have loop anchors within the same proximal locus (Fig. 2a). MAT genes served as housekeeping gene controls in these populations of cells. We then profiled all the called SS18::SSX ChIP-seq peaks (q-value < 0.001) that intersected any TSS in the genome for peak width and enrichment score. Two very distinct patterns emerged among promoter associated peaks: broad peaks, among which the SAT genes were included, and narrow peaks, some of which had very strong fusion enrichment, and among which, MAT genes were included (Fig. 2b). This immediately suggested that SS18::SSX targets promoters in two patterns, narrow and broad, with broad promoter binding designating genes whose transcription was activated across the time course of fusion-driven transformation.

In order to characterize the epigenetic attributes of SS18::SSX-occupied SAT gene promoters, we next profiled the distribution of specific histone marks by ChIP-seq (Fig. 2c-d, Supplementary Fig. 1e). Specifically, in addition to the histone marks introduced above, tri-methylation of lysine 36 on histone H3 was used as a control to mark actively transcribed gene bodies, as opposed to proximal or distal regulatory elements. These ChIP-seq experiments for histone marks revealed a pattern of strong H2AK119ub and H3K4me3 – but curiously lacking H3K27me3 – supporting the observation that these typically silent/poised/bivalent developmental gene promoters are converted towards activated transcription, identified in human and mouse SySs[29,31].

We also sought to characterize the distal (enhancer) binding sites of the fusion that contributed to the regulation of sarcomagenic transcription. Although most loop anchors distant from the promoters of SAT genes colocalized proximally near the TSSs of other genes, others arose from loci distal to all promoters. These distal loci were both broad in width and had strong fusion enrichment (Fig. 2e).

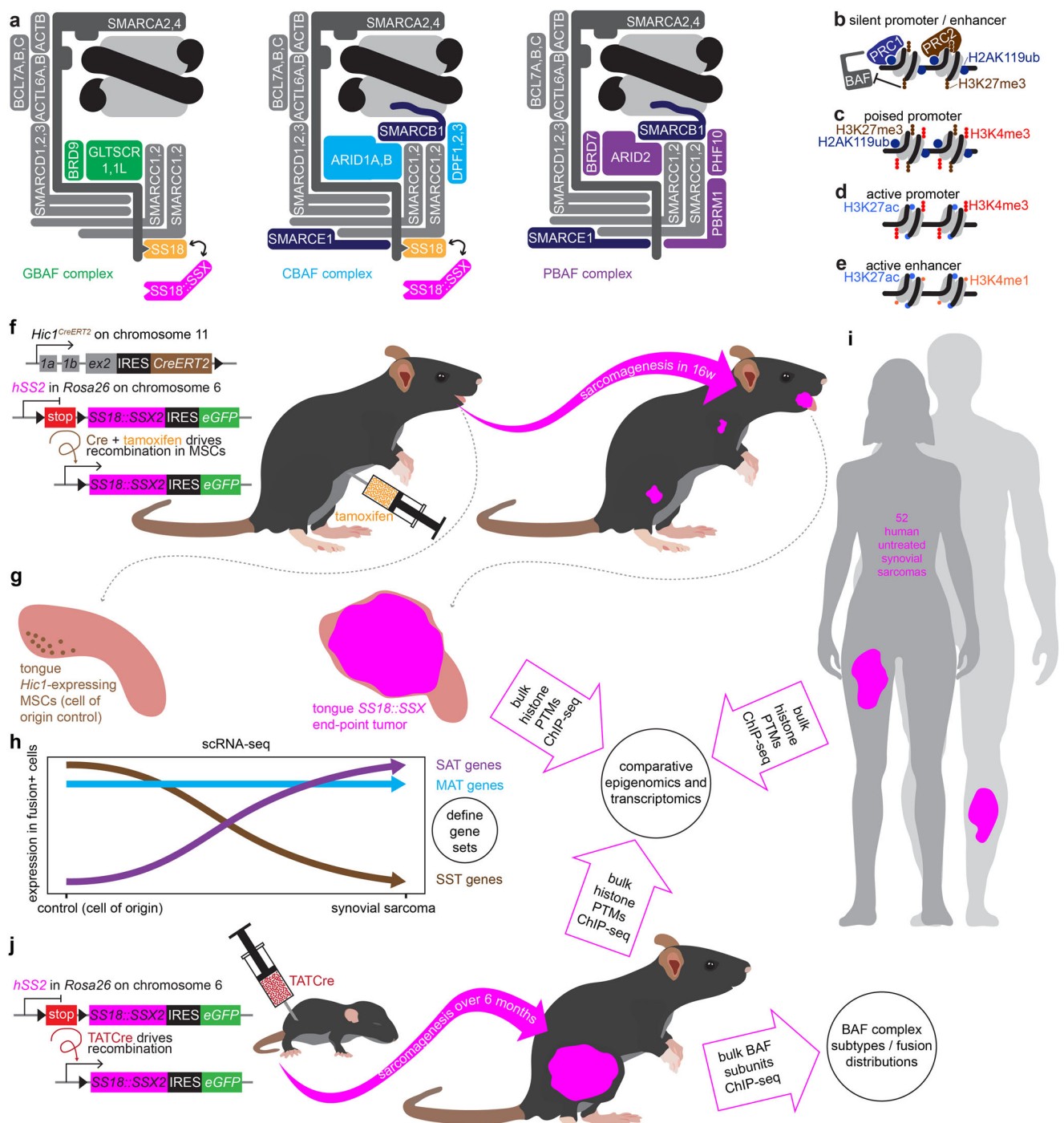

**Fig. 1 | SS18::SSX incorporates into GBAF and CBAF complexes and drives specific transcriptional signatures and epigenomic patterns across a range of mammalian contexts. a** Schematic of the three BAF complex subtypes, non-canonical GLTSCR1-containing (GBAF), canonical BAF (CBAF), polybromo BAF (PBAF). **b** Schematic of two nucleosomes with transcription silencing histone marks, placed by polycomb repressive complexes, PRC1 and PRC2. **c** Histone marks of poised but not expressed bivalent promoters, **d** active promoters, and **e** active distal enhancers. **f** Schematic of the *Hic1^CreERT2^* knock-in allele, which expresses tamoxifen-inducible Cre-recombinase in mesenchymal stromal cells (MSCs), and the *hSS2* allele in *Rosa26*, from which the SS18::SSX2 cDNA is expressed followed Cre-mediated recombination of a floxed stop cassette. Mice injected intraperitoneally with tamoxifen at 6 weeks age develop tumors in a variety of anatomic locations, but most consistently in the tongue over 16 weeks. **g** *Hic1*+ cells isolated from the tongues of 6 week old mice provide a cell of origin control comparison for predictable tongue SyS cells, rendering **h** lists of genes whose transcription is activated by sarcomagenesis (SAT), silenced by sarcomagenesis (SST), or maintained between control cells and SyS cells (MAT). These tongue tumors also provide material for bulk epigenomics that can be compared with **i** human SySs for validation. **j** Mice bearing the *hSS2* allele injected at 8 days of life in the hindlimb with TATCre develop larger tumors that provide the material for epigenomics assessments as well as BAF component ChIP-seq.

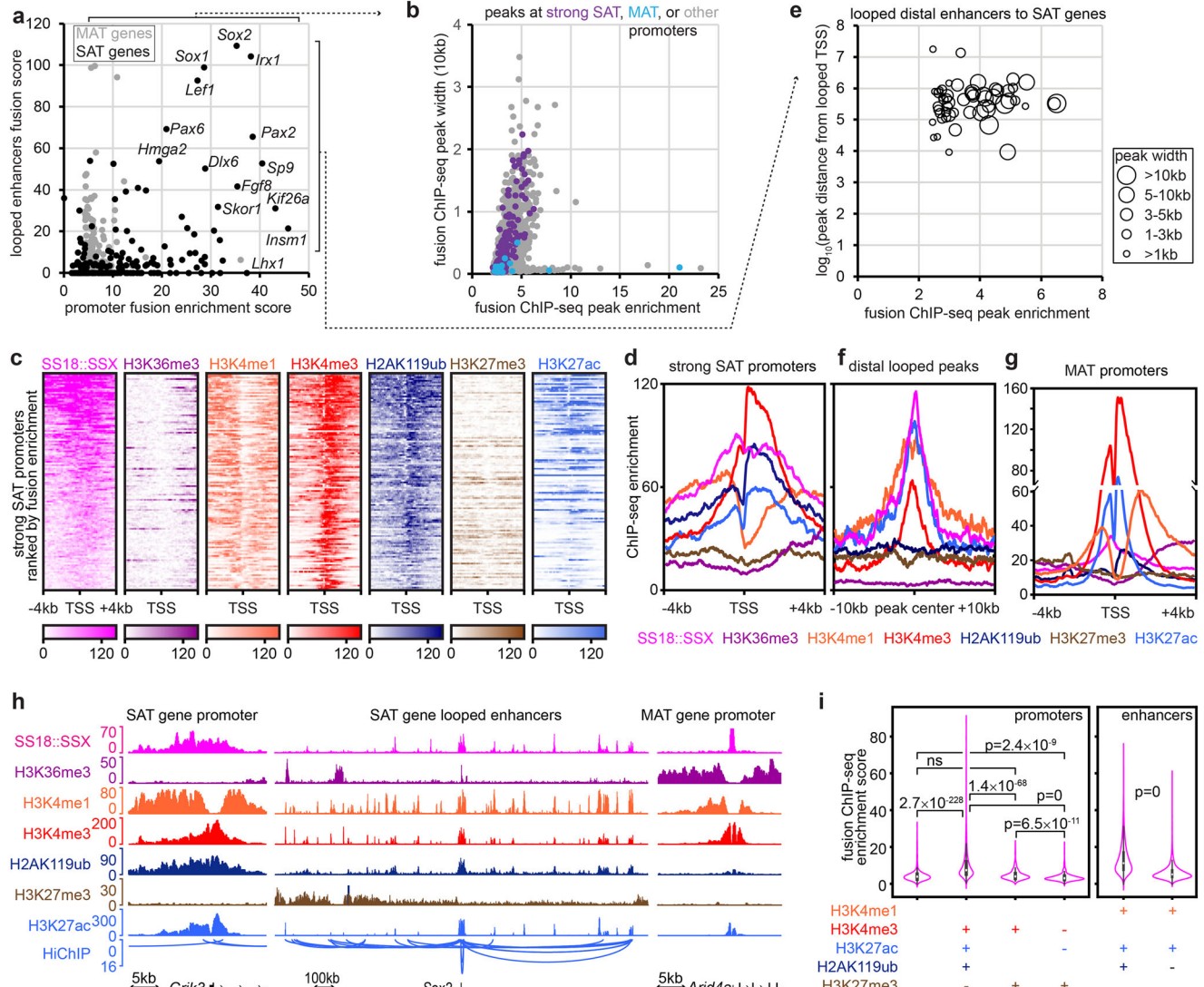

**Fig. 2 | SS18::SSX distributes to transcriptional regulatory elements with monoubiquitylated histone H2A bearing nucleosomes. a** Fusion ChIP-seq summed enrichment at enhancer loci (distal anchors ±2 kb of H3K27ac-HiChIP loop to selected promoters, with enrichment multiplied by loop score) versus promoter enrichment (transcription start site, TSS ± 2 kb) for genes discretely associated with sarcomagenesis activated transcription (SAT) and maintained active transcription (MAT). **b** Fusion ChIP-seq peak width versus enrichment for all peaks that are within 2 kb of any TSS, with those near MAT or strong SAT gene promoters indicated. **c** Heatmaps for ChIP-seq enrichment centered at the TSS of each strong SAT gene promoter. **d** ChIP-seq enrichment plots for histone marks at strong SAT gene promoters. **e** Distal fusion ChIP-seq peak width and distance from its looped SAT gene TSS versus enrichment. **f** ChIP-seq enrichment plots at distal fusion peaks looped to SAT gene promoters. **g** ChIP-seq enrichment plots for MAT gene promoters. **h** Example tracks of a histone promoter signature for an SAT gene promoter and looped enhancers, as well as an MAT gene promoter. **i** Violin plots of fusion enrichment at promoters and enhancers with different patterns of intersected peaks called for histone marks or randomly selected promoters across the genome. Any column missing a plus or minus was simply not specified for ChIP-seq in that category (Violin plots show data distribution with embedded box plots indicating median, 25th–75th percentiles, and whiskers for minimum and maximum. Sample sizes by types of loci: triple positive, $n = 3381$; random, $n = 977$; negative, $n = 940$; bivalent, $n = 325$. Kruskal–Wallis test for unbalanced data, rank-based). Source data are provided as a Source Data file.

H2AK119ub was present at a few of these distal loci (Fig. 2f, Supplementary Fig. 1e). However, in striking contrast, the MAT and SST gene promoters had very different patterns of associated histone mark ChIP-seq enrichment, as they lacked significant and strong H2AK119ub enrichment (Fig. 2h, Supplementary Fig. 1f-h).

We next interrogated these signature histone ChIP-seq enrichment patterns across the entire genome to identify other promoters and distal loci that might interact with the fusion with respect to sarcomagenesis. Promoters were therefore defined by the intersection of called peaks (q-value < 0.001) for this SAT pattern of H2AK119ub +, H3K4me3 +, H3K27ac +, H3K27me3− marks, by the classic bivalent promoter signature (H3K4me3 +, H3K27me3 +), by H3K27me3 alone, or as random promoters across the genome as a control. We

also defined distal active enhancers by called intersection peaks for H3K4me1 and H3K27ac, then divided these between those with or without additionally called peaks for H2AK119ub. SS18::SSX ChIP-seq enrichment at promoters and distal enhancers that included called peaks for H2AK119ub were significantly stronger than the comparison controls (Fig. 2i, Supplementary Fig. 1i-o). Thus, histone mark and fusion ChIP-seq enrichment profiles at these SAT pattern-defined promoters and distal enhancers identified strong coincidence of H2AK119ub and the fusion, consistent with SSX tail-H2AK119ub interaction as the dominant mode of fusion targeting. We next confirmed the pattern of broad fusion promoter peaks coinciding with H2AK119ub and the few narrow promoter peaks less so, by performing ChIP-seq for SS18::SSX and H2AK119ub on two human patient SySs that

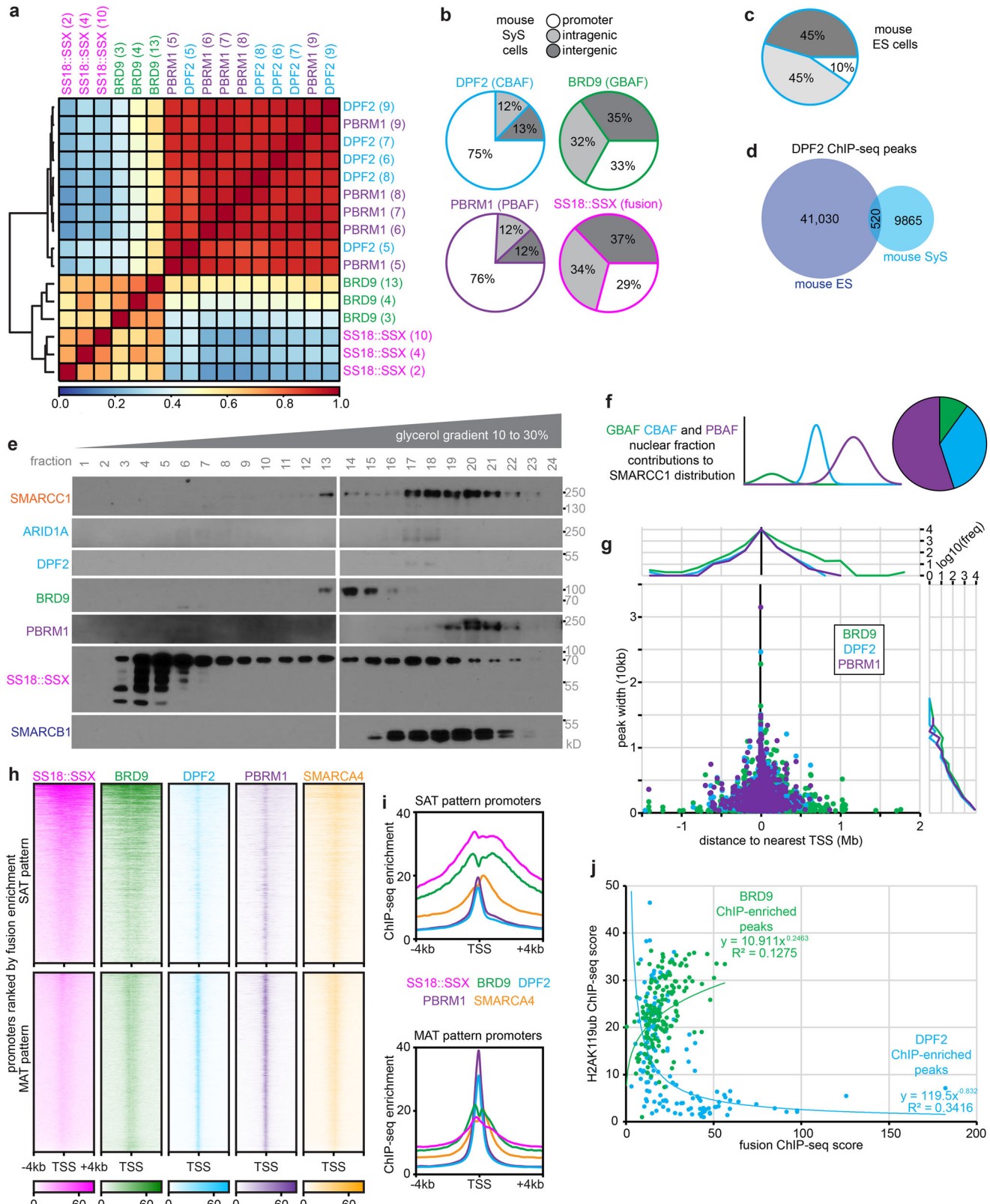

had been expanded as patient derived xenografts prior to whole tumor epigenomics (Supplementary Fig. 1p-r).

**GBAF with fusion distributes to broad proximal and distal regions; CBAF and PBAF distribute narrowly to active TSSs**

The distribution of each BAF family subtype across chromatin was interrogated using ChIP-seq with antibodies against BAF subunits.

Single crosslinking was sufficient for BRD9 (a GBAF subunit), SMARCA4 (also known as BRG1, the principal ATPase subunit common to most BAF-family subtypes), and SMARCC1 (also known as BAF155, a subunit incorporating into all BAF-family subtypes), while PBRM1 (a PBAF subunit) and DPF2 (a CBAF subunit) required double cross-linking, and ARID1A (another CBAF subunit), did not achieve strong, reproducible ChIP-seq with single or double cross-linking

**Fig. 3 | BAF family complex subtypes distribute across chromatin relative to SS18::SSX. a** Correlation heatmap for genome-wide distributions by ChIP-seq of fusion, BRD9, PBRM1, and DPF2 for individual tumors tested. **b** Chromatin annotation discrepancy between CBAF, PBAF, GBAF, and the fusion. **c** Distribution of chromatin annotations for DPF2 ChIP-seq in mouse embryonic stem cells from Zhang et al., 2019. **d** Venn diagram of overlap for DPF2 ChIP-seq peaks in mouse embryonic stem cells (ES)[34], versus mouse synovial sarcomas (SyS). **e** Western blots following glycerol size fractionation of nuclear extracts in a mouse SyS tumor **f** showing contributions of GBAF, CBAF, and PBAF to the overall SMARCC1

distribution calculated for each fraction (This was repeated on two biological replicates, presenting the aggregate clearest blots for one sample). **g** Called peaks width versus distance and histograms from TSS for BRD9, DPF2, and PBRM1. **h** Enrichment heatmaps for SS18::SSX, BRD9, DPF2, PBRM1, and SMARCA4 ChIP-seqs at SAT and MAT pattern promoters. **i** Enrichment plots for SAT and MAT pattern promoters. **j** Plot of the ChIP-seq enrichment scores for fusion versus H2AK119ub, noted for those peaks that are in the top 1% of DPF2 enrichment or BRD9 enrichment, showing divergent relationships in these two populations of peaks. Source data are provided as a Source Data file.

(Supplementary Fig. 2a-b). Strong concordance among biological replicates was confirmed for each antibody. Importantly, strong correlations were found between PBAF and CBAF distributions, as well as between fusion and GBAF distributions (Fig. 3a, Supplementary Fig. 2b-c). The stronger correlation of fusion with GBAF than with CBAF ($r = 0.73$, $r = 0.32$, Pearson correlation coefficients, respectively) was reflected in strikingly distinct annotation distributions between CBAF/PBAF and GBAF/SS18:SSX. CBAF distributes heavily to promoter regions in SyS, contrasting its distribution in murine stem cells, wherein CBAF predominantly distributes to distal intragenic or intergenic loci (Fig. 3b-c)[34]. Notably, CBAF in SyS experienced a near-total depletion at distal loci, positions where it typically plays a vital role in remodeling chromatin at enhancers across various cell types, including stem cells (Fig. 3d)[34].

As previously reported, we confirmed a profound loss of CBAF levels in SyS, using glycerol gradient fractionation of nuclear extracts (Fig. 3e-f)[7]. The relative abundance of GBAF, CBAF, and PBAF in nuclear extracts must be interpreted with the understanding that these assays measure proteins/levels derived from the nucleoplasm, not protein tightly bound to insoluble chromatin. Specifically, the preponderance of SS18::SSX-bearing GBAF (GBAF$^{SS18::SSX}$) that is avidly bound to insoluble chromatin underlies GBAF's underrepresentation in the non-chromatin-bound nucleoplasm that is resolved by the glycerol gradient. Notably, the CBAF-specific subunits DPF2 and ARID1A are difficult to identify in SyS nuclear protein gradients, but are not enriched in the chromatin bound fraction, as we previously reported[7].

GBAF peaks were broader than CBAF or PBAF peaks ($p = 2.88 \times 10^{-13}$, $p = 0.035$, 2-tailed heteroscedastic t-tests, respectively for GBAF-to-CBAF and GBAF-to-PBAF), especially at TSSs, but also comprised the great majority of distal peaks for all BAF complexes (Fig. 3g). Profiling of BAF components at SAT and MAT pattern promoters identified striking contrasts, with CBAF and PBAF narrowly occupying the TSS of both types of promoters, and stronger fusion and GBAF enrichment at SAT pattern than at MAT pattern (H2AK119ub-lacking) or SST promoters (Fig. 3h-i, Supplementary Fig. 2d-e). Distal loci with H2AK119ub show strong enrichment for H2AK119ub, fusion and GBAF. However, certain loci lacking called H2AK119ub peaks also contained GBAF and fusion (and even sub-threshold presence of H2AK119ub) in a similar, albeit diminished enrichment pattern (Supplementary Fig. 2f-i). These data suggest that SS18::SSX distributes with GBAF primarily to the regulatory elements of sarcomagenesis-associated genes. The narrow pattern of CBAF at the TSS (often with accompanying fusion), correlates with gaps in the H2AK119ub and H3K4me3 distribution; GBAF instead co-distributes with H2AK119ub in promoter CpG islands, identified loosely by high GC content of these promoter sequences (Supplementary Fig. 2j-k). Hypomethylated CpG islands are strongly correlated with developmental genes for bivalency as we report in our collaborative study in human SyS[29]. In this fashion, H2AK119ub enrichment inversely correlated with CBAF presence among fusion peaks (Fig. 3j). For confirmation in human cell lines, SS18 ChIP-seq that was performed before and after depletion of SS18::SSX in the Aska SyS cell line, as reported in a prior paper demonstrated a more normal annotation distribution of wild-type CBAF in fusion depleted cells (which the SS18 antibody represents in the absence of the fusion), as well as the particular gain of

predominantly distal enrichments in comparison to the presence of SS18::SSX (Supplementary Fig. 2l)[13].

**Synovial sarcoma transcriptional trajectories define gene sets**
In order to profile the transcriptional changes that accompany SyS development along disparate epithelial, mesenchymal, or poorly differentiated trajectories, we performed single cell transcriptome profiling (scRNA-seq) on mouse SyS tumors comparable to those profiled epigenomically (Fig. 1j). The tumors exhibited a variety of histomorphologies, indicated in hematoxylin and eosin tumor sections of tissue immediately adjacent to that submitted for scRNA-seq (Fig. 4a, Supplementary Fig. 3a). Clusters of tumor-infiltrating fibroblasts, endothelial cells, and immune cells were easily identified by their expression signatures (Fig. 4b-c). Expression profiles of mouse SyS cells were checked against published human SyS scRNA-seq using the SyS expression profile published, which demonstrated strong correlation across species (Supplementary Fig. 3b)[12]. Spatial profiling of transcription in additional mouse SyS tumor samples identified reference genes across histologically-specified cell types (confirmed by immunohistochemistry at the protein level for two examples) and permitted assignment of clustered malignant cells to specific categories (Fig. 4d-f, Supplementary Fig. 3c and 4). We then performed a pseudo-time trajectory from the likely earliest transformed cells to those that displayed mesenchymal spindle cell morphology, epithelial cell morphology, and poorly differentiated morphology (Fig. 4g). These trajectories revealed positively associated genes in both the SAT pattern (sarcomagenesis linked) and MAT pattern (housekeeping) gene sets (Fig. 4h, Supplementary Fig. 5).

**Smarcb1 disruption disables the expression of SS18::SSX-driven developmental target genes**
Next, we sought to determine the functional contribution of the narrow CBAF and PBAF localization at TSSs to SyS tumor phenotypes. Both of these BAF sub-families incorporate SMARCB1 and ARID components. We therefore began by interrogating the impact of *Smarcb1* genetic disruption[40] in *hSS2*-driven tumors (Supplementary Fig. 6a-b). We previously demonstrated that *Smarcb1* knockout significantly shortened latency to tumorigenesis and altered cell morphology away from typical SyS features, with a distinct increase in nuclear atypia (Fig. 5a-b)[7]. This suggests that biology associated with SMARCB1 loss is not included in fusion-only-driven oncogenesis. Transcriptional profiling by RNA-seq demonstrated significantly reduced expression of the majority of SAT pattern genes, inconsistent directional change in many MAT pattern genes, and a minimal change in the SST genes, which are already relatively lowly expressed in the *Smarcb1* wildtype tumors, or the pseudotime associated genes (Fig. 5c-f, Supplementary Fig. 6c). We confirmed that the loss of SMARCB1 leads to profound loss of PBAF complexes and restored the preponderance of CBAF complexes (Fig. 5g-h, Supplementary Fig. 6d), further confirming our prior observation that CBAF complexes can readily assemble in the absence of the SMARCB1 subunit[7]. We previously interpreted similar findings as demonstrating that PBAF loss at the TSS was solely responsible for the acute alteration in tumor phenotype. However, considering the tight correspondence of CBAF and PBAF distributions across the SyS genome, we further propose that CBAF fails to compensate PBAF loss at

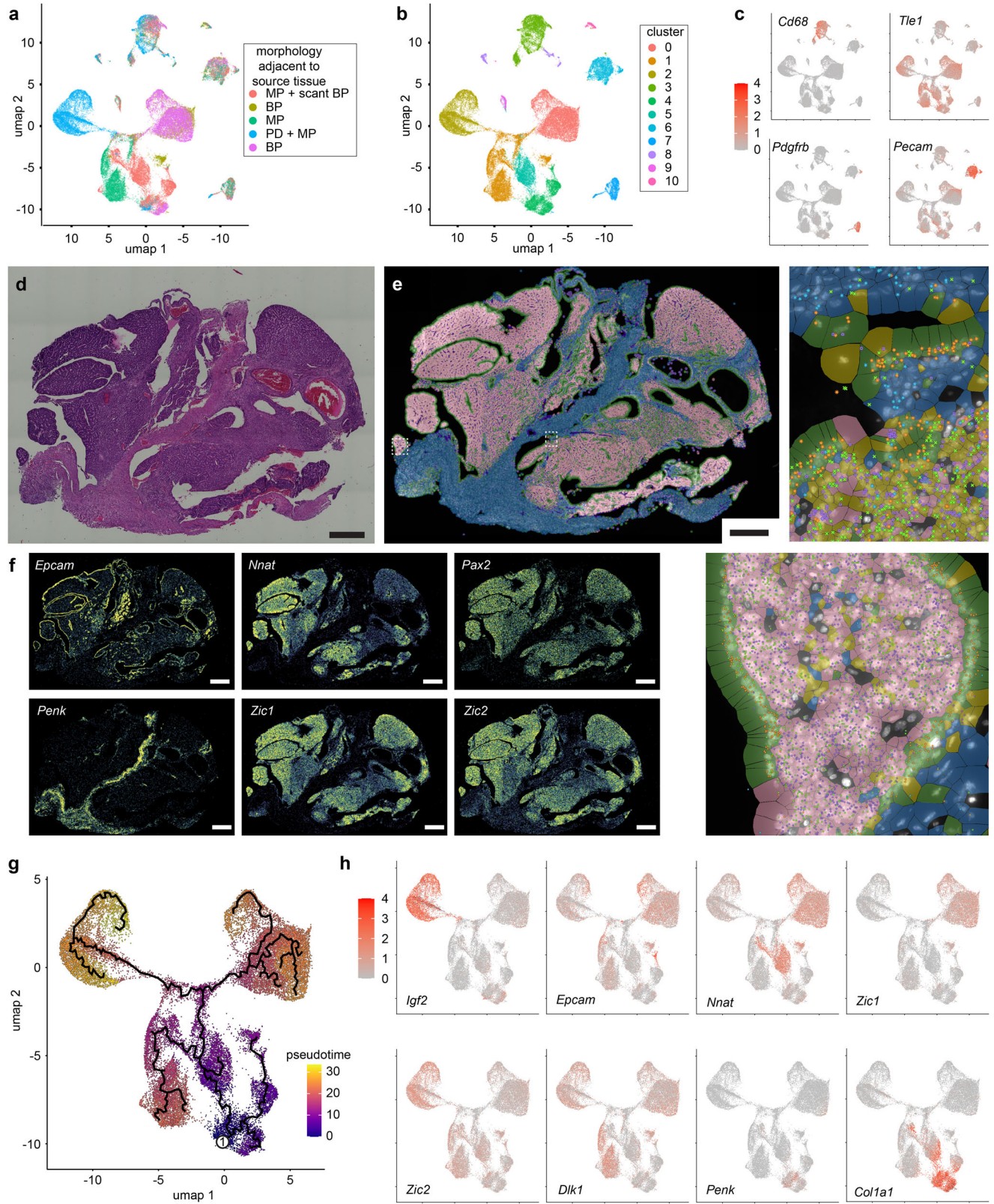

the TSS of SAT pattern gene promoters in the context of SMARCB1 loss. A conspicuous dip at these sites in *Smarcb1*-silenced tumors represents focal loss of general BAF complex ChIP-seq signal here. This dip strongly mirrors that of GBAF distribution in *Smarcb1*-wildtype tumors (Fig. 5i-l). Taken together, in SyS tumors, GBAF abandons the TSS whereas CBAF and PBAF focally occupy the TSS in a SMARCB1-dependent manner.

## *Pbrm1* disruption blocks SS18::SSX reprogramming, encouraging alternate oncogenic pathways

Silencing the PBAF specific subunit *Pbrm1*[41] in *hSS2*-driven tumorigenesis also led to faster tumor development (Fig. 6a, Supplementary Fig. 7a-c) which lacked the full array of SyS histomorphologies observed in *Pbrm1*-wildtype tumors, instead exhibiting increased nuclear atypia (Fig. 6b-c). Compared to the strong impact of *Smarcb1*

**Fig. 4 | Single cell transcriptomics reveal gene sets associated with specific differentiation states of synovial sarcoma cells. a** Seurat clustering by UMAPs in 2 dimensions noting 5 source samples (MP, monophasic; BP, biphasic; PD, poorly differentiated) and clusters **b** of mouse synovial sarcomas initiated by SS18::SSX2 expression by TATCre limb injection at 8 days of life. **c** Heatmaps for expression of marker genes across the clusters, identifying cluster 3 as monocytes/macrophages, 7 as fibroblasts, 6 as endothelial cells, and 0, 1, 2, 4, and 5 as SyS cells, 8, 9, and 10 represent other supporting non-neoplastic cell types. **d** H&E stained photomicrograph of Xenium spatial transcriptomics sample of an hSS2 mouse tumor (all magnification bars are 500 μm; the additional biological replicates of this analysis are presented in Supplementary Fig. 4 in their entirety). **e** Xenium cluster calls for SyS cell types of monophasic (blue), epithelial and poorly differentiated (pink), glandular epithelial cells (green), endothelial cells (purple) with expanded sections demonstrating the positional expression of *Penk* (cyan sunbursts), *Epcam* (orange sunbursts), *Nnat* (purple squares), and *Zic1* (green Xs) transcripts. **f** Xenium heatmap expression data for marker genes of distinct histological cell types. **g** Whole transcriptome scRNA-seq of neoplastic cell clusters demonstrate divergent pseudotime trajectories into the varied Seurat UMAP clusters, beginning at the circled 1. **h** Heatmap expression of individual marker genes that distinguish between different neoplastic cell clusters.

on SAT pattern gene expression in SyS tumors, *Pbrm1* silencing was comparatively subtle (Fig. 6d). This argues that CBAF (with SMARCB1 intact) can compensate for PBAF disruption at the narrow TSS binding positions in SyS, remodeling chromatin to enable gene expression. More striking was the reduced expression of MAT pattern genes in *Pbrm1*-silenced tumors (compared to *Smarcb1* silencing) (Fig. 6e). *Pbrm1* silencing in these hSS2 tumors achieved the anticipated loss of *Pbrm1* transcripts by RNA-seq and the reduction in the presence of PBAF complexes in nuclear extracts (Fig. 6f-h). Because CBAF and PBAF ChIP-seq enrichments are strongly correlated across the entire genome (Fig. 6i), the modest (relative to *Smarcb1* disruption) differential expression in the context of *Pbrm1* disruption argues that CBAF can compensate for PBAF loss at the TSSs of many expressed genes.

For optimal rigor, we also performed the experiments in the hSS1 background, expressing SS18::SSX1 from *Rosa26* via TATCre injection, which drives slightly slower synovial sarcomagenesis[6]. While *Pbrm1* loss decreased latency to tumorigenesis in hSS2 tumors, homozygosity for floxed *Pbrm1* in hSS1 tumors paradoxically slowed tumorigenesis (Fig. 6j, Supplementary Fig. 7d-f). Tumors arising more slowly in *hSS1;Pbrm1-fl/fl* mice retained normal SyS histomorphologies and lacked the nuclear atypia seen in *hSS2;Pbrm1-fl/fl* (Fig. 6k). Tumors developing in *hSS1;Pbrm1-fl/fl* mice also depleted *Pbrm1* expression (by RNA-seq) less than did hSS2 tumors (Supplementary Fig. 7g, see also Fig. 6f), suggesting that tumor cells recombining only one of the two floxed *Pbrm1* alleles experienced stronger positive selection in the hSS1 background. Principal component analysis (PCA) of entire transcriptomes determined by RNA-seq of *Pbrm1-fl/fl* and *wt/wt* tumors from both hSS1 and hSS2 mice demonstrated disparate clustering of the *hSS2;Pbrm1-fl/fl* tumors, but retained clustering of most *hSS1;Pbrm1-fl/fl* tumors with *Pbrm1* wildtype tumors from both backgrounds (Fig. 6l). Overall, this apparent paradox suggests that *Pbrm1* silencing similarly blunted synovial sarcomagenesis programs in both SS18::SSX backgrounds. This blunting of SyS programs tipped the faster growing hSS2 tumors[6] into an alternate tumorigenesis program, but merely slowed hSS1 tumor development, causing selection for alleles that escaped recombination. The expression of hSS1 and hSS2 tumors has been tested previously, but again did not show significant alterations in important pathways, only a shift in proliferation and latency (Supplementary Fig. 7h)[6].

### *Arid1a* or *Arid1b* disruption enhances the transcriptional silencing derived from SS18::SSX expression

Silencing of *Arid1a* or *Arid1b* sped tumorigenesis in both backgrounds (insignificantly in hSS2/*Arid1b*-loss mice, but significantly in all others), without significant change in the histomorphology of the developing tumors (Fig. 7a-f, Supplementary Fig. 8a-h)[42,43]. Tumors with homozygous disruption of *Arid1a*, especially, demonstrated rounder nuclei and higher cellular density (Fig. 7e). *Arid1a* or *Arid1b* silenced tumors still exhibited strong epithelial features, with or without epithelial gland formation (Fig. 7f). Similar to *Pbrm1*-silenced hSS2 tumors, more of the SAT pattern genes had higher expression in *Arid1a*-wildtype hSS2 tumors than in tumors with silenced *Arid1a*, but the difference was even more subtle (Fig. 7g). MAT genes were similarly not impacted to any large degree (Fig. 7h), even less so than in *Pbrm1*-silenced hSS2

tumors. More strikingly, with *Arid1a* disruption, the expression of synovial sarcoma transcriptional silenced (SST) genes was further reduced (Fig. 7i-j). Although there was no appreciable further depletion of CBAF complexes in *Arid1a*-silenced tumors beyond that of wildtype hSS2 tumors, there was also no apparent compensatory increase in expression of either paralog upon deletion of the other (Fig. 7k, Supplementary Fig. 8i). Among the genes exhibiting increased expression along the scRNA-seq pseudo-time trajectory toward more reprogrammed states, most of those that were MAT pattern genes had higher expression in *Arid1a*-silenced tumors (Fig. 8a). Overall, *Arid1a* or *Arid1b* disruption rendered tumor transcriptomes that clustered within the hSS2 tumor transcriptomes, unlike *Pbrm1* disrupted tumors (Fig. 8b). Similarly, Xenium assessments of *Pbrm1*-silenced tumors showed a wider departure from the core clusters of neoplastic cell types than an *Arid1a*-silenced tumor (Fig. 8c, Supplementary Fig. 9). Having firmly established the expression of SAT gene signatures in human SyS in our related manuscripts[29,31], we next sought to confirm that these CBAF-loss related signatures of gene expression and repression are also conserved between species with SS18::SSX-driven malignancy. The SST genes that were silenced more profoundly by *Arid1a* genetic disruption were found to be very lowly expressed in 52 human SyS transcriptomes. Inversely, the pseudo-time trajectory associated MAT pattern genes that were further upregulated by *Arid1a* silencing were highly expressed in the same human SyS tumors and their elevated expression correlated with worse prognosis, as well (Fig. 8d, Supplementary Fig. 10a).

### Discussion

Like SS18, the SS18::SSX fusion incorporates into both GBAF and CBAF (but not PBAF), and is found coincident with both subfamilies of BAF complexes on chromatin—all of which was predicted by biochemical and cell line based experiments by ourselves and others previously. Here, we have learned in the in vivo context that SS18::SSX has distinct relationships at gene regulatory elements bearing specific epigenomic patterns with each of these complexes.

Fusion oncoprotein distribution follows H2AK119ub patterns, displaying broad peaks both within and outside promoters, and appears to be the driving force behind GBAF distribution (Fig. 8e). Following fusion expression, this gain of targeting and functional activity for GBAF complexes involves promoters (and distal elements) that transition from bivalency in progenitor cells to active histone marks in the tumor cells[31], likely in part through chromatin opening by GBAF. The changing patterns of histone marks over time and tumor development were demonstrated through comparisons with a known cell of origin in our collaborative mouse experiments (Fig. 1 f-h)[31]. The particular patterns we identified across a series of developmental target genes supports interpretations of the human profiles of SyS tumors that exhibit variable retention of bivalency for H3K27me3 and H3K4me3 versus complete conversion to H3K4me3 coincident with H2AK119ub occupancy, though curiously lacking H3K27me3 (Fig. 1i)[29]. Here, we demonstrate that the strong relationship between H2AK119ub and H3K4me3 is at least coincident—if not driven by—the presence of GBAF (with incorporated SS18::SSX, Fig. 3). These genes, that we designated as SAT pattern genes, comprise the direct target

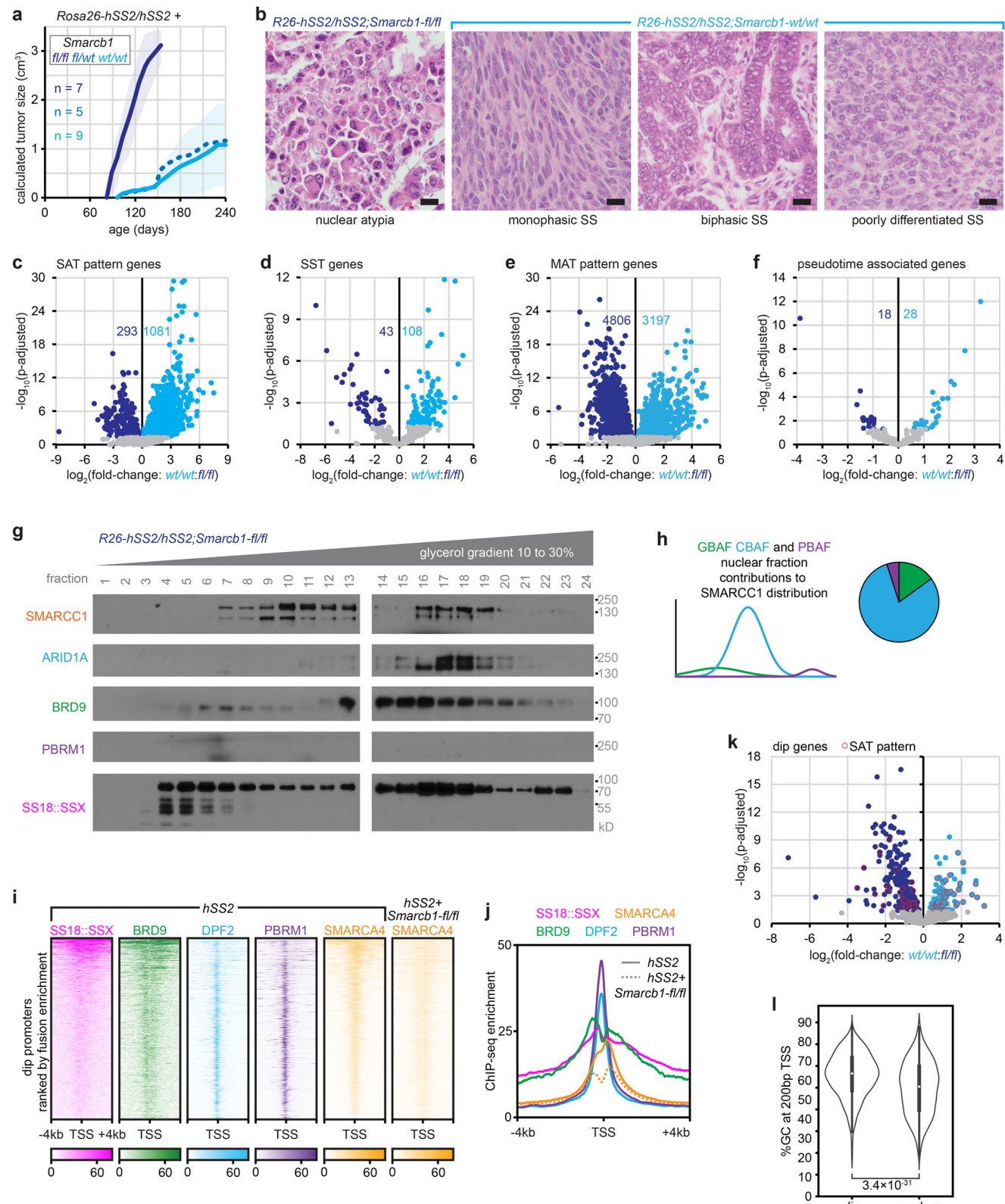

genes of the fusion oncoprotein, which are driven toward elevated transcription during transformation. Gene targets of GBAF^SS18::SSX comprise an activated expression signature that is strongly shared across species (human and mouse)[29,31]. As much as these SAT pattern genes are targeted by GBAF^SS18::SSX, their expression also depends on the functional presence of either PBAF or CBAF (each incorporating SMARCB1) narrowly at the TSS. Whereas GBAF is apparently retained at

SAT promoters in the absence of SMARCB1, *Smarcb1* silencing causes CBAF and PBAF to abandon the TSS, which is coincident with transcriptional attenuation and loss of SyS identity (Fig. 5i-j). This finding suggests that GBAF^SS18::SSX has limited capacity to bind to and remodel chromatin at the TSS itself, and instead may open the region for the action of transcription factors, along with CBAF and PBAF, ultimately to activate the genes. Therefore, target genes that are transcriptionally

**Fig. 5 | SMARCB1 loss limits PBAF assembly in SyS and blocks CBAF from TSS binding. a** Growth trajectories of mean tumor size (sample sizes indicated are biological replicates: individual tumors developing in individual mice, where shading indicates ± standard deviations, p-values from two-tailed, heteroscedastic *t* tests) in *hSS2* mice injected with TATCre at day 28 of life, comparing littermates with varied *Smarcb1-fl* genotypes. **b** Example photomicrographs of the bizarre nuclear atypia apparent in an *hSS2* tumor with homozygous floxed *Smarcb1* and the three typical categories of SyS histology identified in *Smarcb1* wildtype tumors (An H&E stained section of each of the tumors produced was reviewed at the same sample size as the tumorigenesis experiments in (**a**) but representative photomicrographs were procured secondarily; each magnification bar is 10 μm). **c** Differential expression (by bulk tumor RNA-seq) of SAT pattern genes, **d** SST genes, **e** MAT pattern genes, and **f** the genes associated with the pseudo-time trajectory noted in scRNA-seq (see Fig. 3g) of *hSS2* tumors bearing *wildtype* versus homozygous floxed *Smarcb1*. **g** Western blots and **h** graphical histograms indicating the relative contribution of GBAF, CBAF and PBAF in glycerol nuclear fractions of an *hSS2, Smarcb1*-disrupted tumor (This experiment was repeated on two biological replicates, presenting the aggregate clearest blots for one sample). **i** Enrichment heatmaps and **j** enrichment plots for BAF components ChIP-seq at promoters defined as having focal loss (dip) of SMARCA4 in *hSS2;Smarcb1-fl/fl* tumors. **k** Differential expression of genes defined by the dip at the promoter comparing *hSS2;Smarcb1-wt/wt* and *hSS2;Smarcb1-fl/fl* tumors. **l** Violin plots of the GC content of the 200 bp surrounding the TSS with dip genes or random other promoters across the genome (Violin plots show data distribution with embedded box plots indicating median, 25th–75th percentiles, and whiskers for minimum and maximum. Sample sizes: dip sites, $n = 1217$; random sites, $n = 977$; Kruskal-Wallis test for unbalanced data, rank-based; p-value from Dunn test for pairwise comparisons). Source data are provided as a Source Data file.

activated by the fusion oncoprotein are marked broadly by H2AK119ub, H3K4me3, and GBAF, but additionally require either PBAF or CBAF narrowly at the TSS.

In contrast to GBAF, the distribution of the residual CBAF[SS18::SSX] complexes in SyS appears to be directed primarily by CBAF itself (or interacting transcription factors), with the fusion accompanying as a passenger. Consistently, CBAF[SS18::SSX] is found in narrow peaks near TSSs of promoters lacking H2AK119ub altogether, or at least lacking H2AK119ub focally near TSSs (Fig. 3). This is not the normal distribution of CBAF, as the typically-distal enhancer positions of CBAF are almost completely abrogated in SyS. We cannot interpret these data to indicate that CBAF[SS18::SSX] is not recruited to H2AK119ub sites—biochemical experiments have clearly demonstrated that it is—only that it does not remain there in enriched abundance at steady state. There are still some very broad H2AK119ub-decorated loci with marked abundance of all the BAF family subtypes, such as the *Hoxb* cluster (Supplementary Fig. 2j), but most CBAF sites in the SyS in vivo context avoid H2AK119ub, at least focally. Curiously, this focal localization of CBAF and PBAF at the TSS requires SMARCB1. Here, future work will determine how SMARCB1 helps CBAF and PBAF recognize histone determinants at active gene TSSs in SyS, but potentially conflicts with H2AK119ub.

The overall reduction in CBAF complex protein levels directed by SS18::SSX is considerable, but incomplete[7], as a small—but still significant—fraction of CBAF complexes remains, usually bearing the fusion oncoprotein SS18::SSX and SMARCB1. Importantly, we report the near complete redistribution of these CBAF complexes on chromatin. The observation that disruption of *Arid1a*, which codes for a critical subunit of CBAF, reduced the latency to tumorigenesis without significantly altering tumor phenotype, argues that CBAF-dysfunction only furthers or enhances aspects of the tumorigenesis process that are otherwise intrinsic to SS18::SSX expression. While ARID1B might compensate for ARID1A absence in these cells, as these are paralogous members of CBAF and ARID1B blunts the effect of ARID1A loss in biochemical assembly experiments, the loss of ARID1B had an even smaller effect on tumors. There was no noticeable change in the expression of either component when the other was silenced, casting some doubt on any compensatory role. Considered in total, each of these individual genes is a powerful single-gene tumor suppressor across many types of cancer[44,45]. That even the singular loss of each has such little impact on the phenotype of SyS—beyond reducing latency to tumorigenesis—suggests that the CBAF complex in which they function is important mostly to SyS by its incident dysfunction.

The key observation of the varied impacts of *Arid1a*, *Arid1b*, *Pbrm1*, and *Smarcb1* disruptions on synovial sarcomagenesis in the mouse cannot be observed in other readily available experimental platforms. Although disruptions of their homologous genes are each observed in human SyS, and at somewhat elevated preponderance in the population of human SySs compared to other soft-tissue sarcomas

(Supplementary Fig. 10b), we cannot understand their contribution to the speed of tumorigenesis or even in the retention of true transcriptional phenotypes from observed single cases in humans, which were diagnosed not by a strict pattern of histology or transcriptome, but by the presence of the SS18::SSX fusion itself. Further, the dependency (on *Smarcb1* and *Pbrm1*) for true synovial sarcomagenesis and lack thereof (on *Arid1a* and *Arid1b*) would not be reflected in a dependency such as has been explored in the RNA interference and CRISPR-mediated single gene depletions reported by the DepMap[46] (Supplementary Fig. 10c), which only reflect gene dependencies for basic proliferation rates or cell survival in vitro.

Taken together, our in vivo work reveals two main and distinct mechanisms for SS18::SSX-mediated transcriptional reprogramming via BAF complexes in SyS: 1) upregulation of gene targets of GBAF with SS18::SSX, that bear H2AK119ub in their promoters and looped enhancers, and 2) silencing/attenuation of CBAF targets due to diminished and redistributed CBAF. Here, transcriptional silencing likely involves loss of typical CBAF distributions to distal regulatory elements (enhancers), which rely on ARID1A (thus, CBAF distribution losses from fusion expression are amplified by *Arid1a* silencing). As the impact of the fusion is pronounced, the effect of CBAF dysregulation within cells expressing the fusion may vary across different cells of origin and possibly contribute heavily to the cell death phenotype driven by SS18::SSX expression in many cell types[6]. This may be attributable to the toxicity that is typically associated with BAF disruptions, and raises the interesting question regarding how cell context shapes the in vivo survival and adaptation of mesenchymal stromal cells to SS18::SSX fusion expression. Importantly, both the CBAF-loss silenced genes and a proposed list of CBAF-loss upregulated genes lacking typical features of Polycomb targeting, show highly conserved trajectories of differential expression among human SySs and correlate with prognosis (Fig. 8d, Supplementary Fig. 10a). We propose that these CBAF-dysfunction associated changes are another important aspect of the transcriptional reprogramming that accompanies expression of SS18::SSX. As future therapeutic efforts variably target each of these mechanisms (gain of GBAF function at polycomb targets or loss of CBAF function), the gene sets we have identified as associated with each constitute vital datasets to interrogate mechanisms and efficacies of therapeutics.

Notably, any additional genetic perturbation added to SS18::SSX expression led to at least slight reductions in the expression of SAT pattern genes. Although profoundly reduced expression of these genes in tumors lacking *Smarcb1* may be explained by loss of PBAF and CBAF at the TSS (where GBAF[SS18::SSX] may be incapable of binding), the disruptions of either PBAF or CBAF by other subunit silencing experiments also diminished expression of subsets of SAT pattern genes at least slightly. This observation fits a model for heterogeneity in human SyS, where tumors that have more genomic copy number variation also have more retained bivalency, as shown in our human datasets[29]. It may be that CBAF dysregulation (or alternative oncogenic

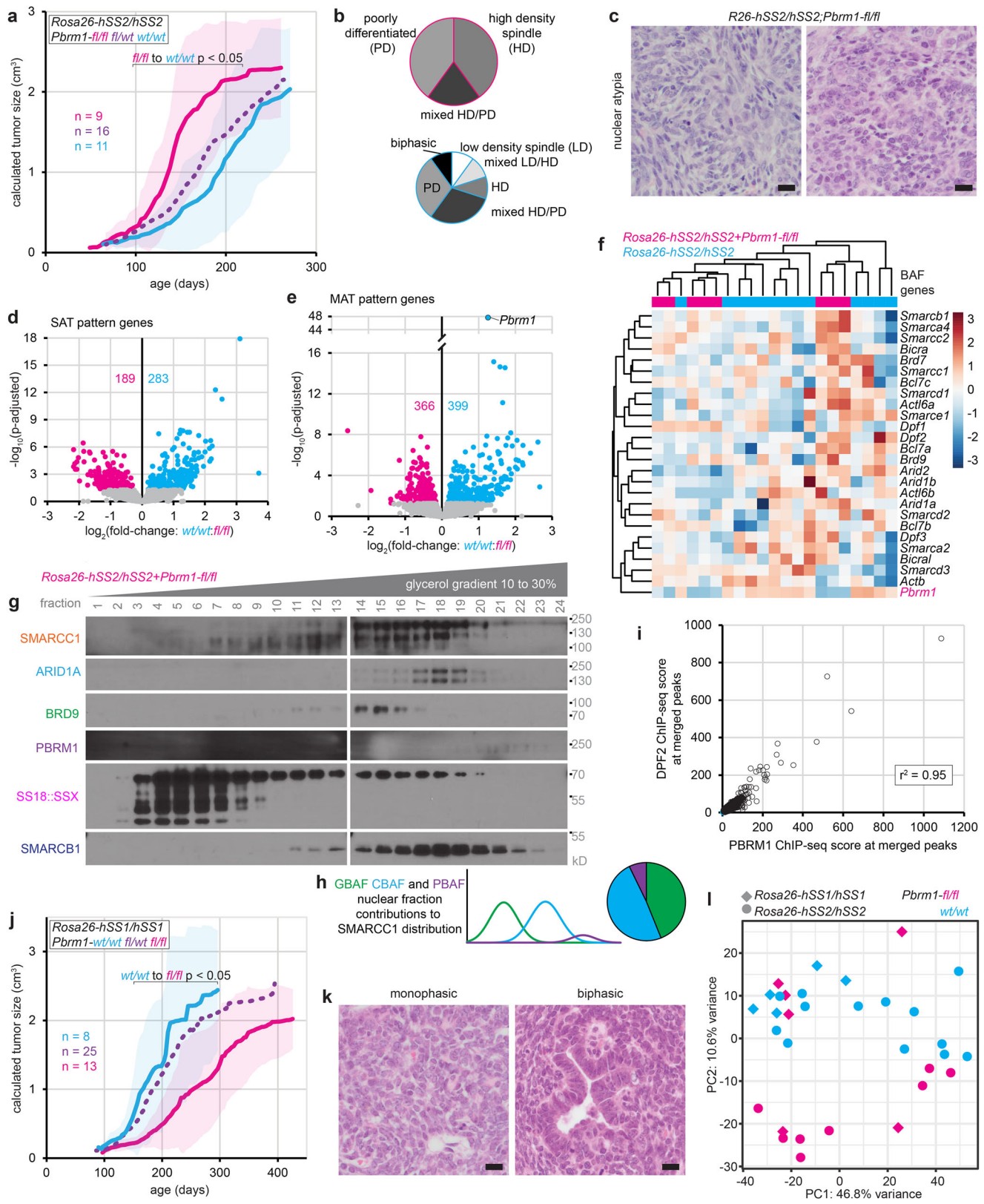

mechanisms) in SyS reduce the dependence on GBAF$^{SS18::SSX}$ for complete reprogramming at developmental genes, since onco-transformation may also be fundamentally enabled by CBAF-dysfunction-induced transcriptional dysregulation.

Most of the efforts to develop SyS-directed epigenetic therapies have targeted biology that has been identified during the almost decade-long pursuit of the mechanism of gained transcriptional

activity by the fusion. These have included bifunctional degraders of BRD9, a member of GBAF, and WDR5, which places the H3K4me3 histone mark that is amplified at SAT genes[20,26,47]. The impact of the first of these was modest at best and produced unexpected toxicities in clinical trials[48,49], both of which phenomena have yet to be fully explained mechanistically. DNA methyltransferase inhibition was pursued because DNA hypomethylation determines the genome-wide

**Fig. 6 | PBRM1 loss prohibits PBAF assembly and blocks the synovial sarcoma phenotype. a** Growth trajectories of mean tumor size (sample sizes indicated are biological replicates: individual tumors developing in individual mice; shading indicates ± standard deviations; p-values from two-tailed, heteroscedastic *t* tests, with individual p-values listed in Source Data file.) in *hSS2* mice injected with TATCre at day 8 of life comparing littermates with varied *Pbrm1-fl* genotypes. **b** Pie charts indicating the histomorphology of *hSS2;Pbrm1-fl/fl* versus *hSS2;Pbrm1-wt/wt* tumors (*n* = 15, *n* = 10, biological replicates of individual tumors). **c** Example photomicrographs of the modest nuclear atypia noted in *hSS2;Pbrm1-fl/fl* (An H&E stained section of each of the tumors produced was reviewed at the same sample size as in (**b**) but representative photomicrographs were procured secondarily; each magnification bar is 10 μm). **d** Differential expression (by bulk tumor RNA-seq) of SAT pattern genes and **e** MAT pattern genes of *hSS2* tumors with *wildtype* versus homozygous floxed *Pbrm1*. **f** Expression heatmap of BAF component genes in *hSS2* tumors with or without *Pbrm1* disruption. **g** Western blots for BAF components after glycerol gradients of nuclear fractions of a *hSS2;Pbrm1-fl/fl* tumor (This experiment was repeated on two biological replicates, presenting the aggregate clearest blots for one sample). **h** Graphical histograms indicating the relative contribution of GBAF, CBAF, and PBAF to SMARCC1 distributions in an *hSS2, Pbrm1*-disrupted tumor. **i** DPF2 versus PBRM1 ChIP-seq enrichment across merged peaks between the two. **j** Growth trajectories of mean tumor size in *hSS1* mice littermates with varied *Pbrm1-fl* genotypes (sample sizes indicated are biological replicates: individual tumors developing in individual mice; shading indicates ± standard deviations; p-values from two-tailed, heteroscedastic *t* tests, with individual p-values listed in Source Data file.) **k** Photomicrograph examples demonstrating retained SyS phenotypes without nuclear atypia in the *hSS1;Pbrm1-fl/fl* tumors which develop more slowly than tumors in *hSS2;Pbrm1-fl/fl* mice (An H&E stained section of each of the tumors produced was reviewed at the same sample size as the tumorigenesis experiments in (**j**) but representative photomicrographs were procured secondarily; each magnification bar is 10 μm). **l** PCA plot in two dimensions from bulk RNA-seq of whole tumor transcriptomes in *hSS1;Pbrm1-fl/fl*, and *hSS2;Pbrm1-fl/fl*, mice. Source data are provided as a Source Data file.

distribution of H2AK119ub by PRC1, which strategy was effective preclinically, but prompted more recovery of genes silenced in SyS (SST genes) than reduced activation of SAT genes[50]. The role of BAF remodeling enzymatic activity itself has also been minimally explored, but represents a complex relationship with nuances, as it is not strictly a dependency in human SyS cell lines (Supplementary Fig. 10b-c)[21]. A recent report targets SUMOylation, which leads to the recovery of otherwise depleted CBAF complexes, which may provide additional targetable vulnerabilities as the biology of CBAF dysfunction in SyS is more deeply explored[51].

## Methods

### Ethical approval

The research described herein complies with all relevant ethical regulations. The University of Utah Institutional Animal Care and Use Committee approved all the vertebrate animal experiments. The maximal tumor size permitted in our approved protocols is 10% of the mouse' body mass, which size was not exceeded in our study. Any mouse reaching this size is humanely euthanized, having reached endpoint. Mice were housed at a controlled ambient temperature of 68–78 °F with a relative humidity of 30–70%. Animals were maintained on a 12-hour light/dark cycle, with lights on at 6:00 AM and off at 6:00 PM.

### Animal studies

Tumor growth rates in mice carrying *Arid1a-fl/fl*[43], *Arid1b-fl/fl*[42], or *Pbrm1-fl/fl*[41], on an *hSS2* (*SS18::SSX2*)[5] or *hSS1* (*SS18::SSX1*)[6] background were interrogated. Spontaneous tumors with each combination genotype were generated by limb injection with TATCre protein at 8 days of age. Caliper measurements started when tumors became visible. Tumor volumes were calculated using the following formula: tumor volume = $(D \times d^2)/2$, in which D and d refer to the long and short tumor diameter, respectively. All males and females from experimental litters were included in each group. Genotyping primer sequences are in the Key Resources Table.

### Histology

Time of morbidity was variable across individuals and groups, at which point mice were euthanized and tumors harvested at necropsy. Tissues were fixed in 4 percent paraformaldehyde, dehydrated in increasing ethanol gradients, embedded in paraffin, and sectioned at 10μm thickness for standard H&E staining. For the quantitative assessments of histological features in the different groups, slides were reviewed after randomization for order, blinded regarding the genotype of the mouse from which the tumor was harvested. Nuclear ratios were determined dividing the length of the long axis of each nucleus over the length of the orthogonal axis. These were averaged from three measurements in each histomorphologically distinct area of the tumor section, then multiplied by the fractional contribution of that histomorphological area to the whole section. Nuclear density was calculated for each histomorphologically distinct region as a count of nuclei in a 500 μm² square on the photomicrograph.

### Immunofluorescence

Slides were deparaffinized in Xylene 2 x 5 min then rehydrated in 100% ethanol 2 x 2 min, in 70% ethanol 1 x 2 min then washed in PBS 2 x 5 min. Antigen retrieval was performed by boiling slides in 0.01 M citrate buffer for 10 min then an additional 20 min on the bench top as the citrate buffer cooled. Slides were then washed in PBS 1 x 5 min and incubated in block solution containing 2.5% BSA (Sigma A7030) and 2.5 Goat serum (Gemini 100-190) for 1 hour at room temperature prior to incubation in primary antibody (see antibody list) overnight at 4 °C. Slides were then incubated in Alexa flour conjugated secondary antibodies (Supplementary Table 1) diluted 1:500 for 1 hour at room temperature. After secondary antibody staining, slides were washed 3 x 5 min in PBS and counterstained with DAPI (600 nM) before mounting with Aqua Polymount (Polysciences, Inc. 18606).

### Mouse tumor nuclear extraction

Tumors were pulverized at −80 °C in Covaris TT1 tissue tubes (Covaris). Tissue pellets were washed and lysed with Buffer A (20 mM HEPES pH 8.0, 1.5 mM MgCl2, 10 mM KCl, 0.25% NP-40, 0.5 mM DTT with 2x protease inhibitor cocktail). Pelleted nuclei were then resuspended in Buffer C (20 mM HEPES pH 8.0, 25% glycerol, 1.5 mM MgCl2, 420 mM KCl, 0.25% NP-40, 0.2 mM EDTA, 0.5 mM DTT, and 2x protease inhibitor cocktail) and disrupted using a dounce homogenizer (Sigma).

### Density sedimentation gradients and calculation of BAFs abundances

Nuclear extractions from tumors were diluted 1:1 with dilution buffer (20 mM HEPES pH 8.0, 1.5 mM MgCl2, 0.2 mM EDTA, 0.5 mM DTT, and 2x protease inhibitor cocktail). Samples were loaded onto tubes (Beckman) with 11 ml 10-30% glycerol gradients containing 20 mM HEPES pH 8.0, 1.5 mM MgCl2, 200 mM NaCl, 0.2 mM EDTA, 0.5 mM DTT, and 2x protease inhibitor cocktail. Tubes were then loaded into a SW41 rotor and centrifuged at 40,000 rpm (approximately 280,000 x g) for 20 hours at 4 °C. 24 fractions were automatically collected with a BioComp fractionation system (BioComp), with each fraction containing 466 μL of sample. Fractions were run on an SDS-PAGE gel for Western blot analysis. The western blot were quantified by ImageJ software. The abundances of BAF-family subtypes were normalized across gradient levels for each protein component, represented by gradient fractions 12–24. For each subtype-specific component (BRD9 for GBAF, DPF2 for CBAF, and PBRM1 for PBAF), a normal distribution curve with a specified mean and standard deviation was fitted to the

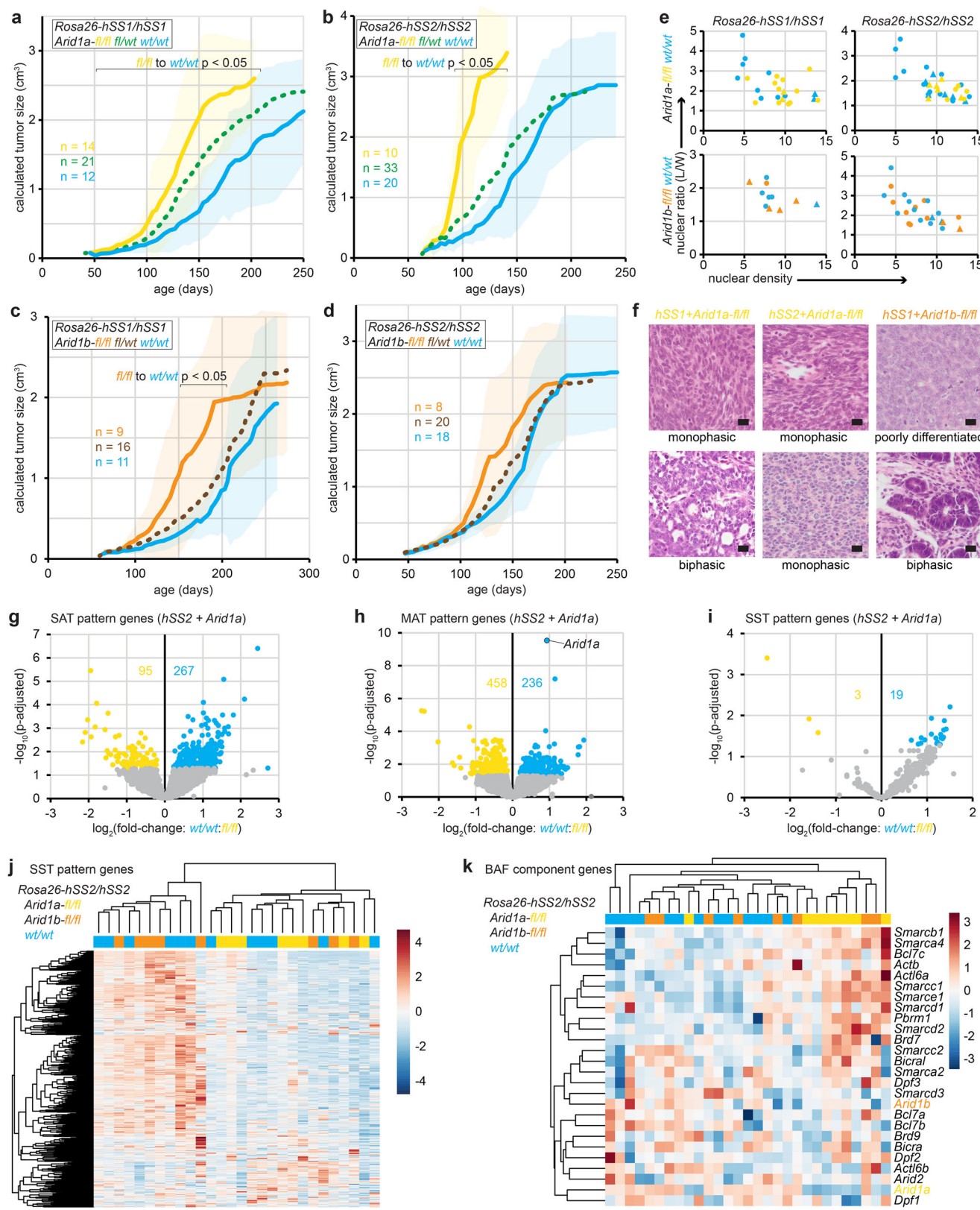

protein's blot data. These distributions were used to calculate the contribution of each BAF subtype to the total BAF complex across gradient fractions by multiplying the fraction of each subtype's complexes by their abundance in that gradient fraction. The total predicted BAF for each gradient fraction was then compared to the observed distribution of SMARCC1 (common to all BAF subtypes), testing various integer percentages of GBAF, CBAF, and PBAF. The optimal fit

minimized the sum of squared differences between predicted and observed BAF distributions across fractions 12–24. The best-fit fractional abundances of GBAF, CBAF, and PBAF were calculated.

## Western blotting

Primary antibodies used for western blots are listed in the Key Resources Table. Tumor protein samples were separated by 10% PAGE

**Fig. 7 | CBAF disruption enhances tumorigenesis without compromising the synovial sarcoma phenotype. a** Growth trajectories of mean tumor size (sample sizes indicated are biological replicates: individual tumors developing in individual mice; shading indicates ± standard deviations, p-values from two-tailed, heteroscedastic *t* tests, with individual p-values listed in Source Data file.) in *hSS1* and **b** *hSS2* mice injected with TATCre at day 8 of life comparing littermates with varied *Arid1a-fl* genotypes. **c** Similar tumor growth curves for *hSS1* and **d** *hSS2* for littermates with varied *Arid1b-fl* genotypes. **e** plots of nuclear shape ratios (closer to 1 indicates roundness) and density for tumors from the different genetic cohorts, with monophasic and poorly differentiated tumors across the spectrum in round dots and tumors with epithelial histomorphology indicated by triangles. **f** Example photomicrographs of tumors with retained SyS histomorphological features following varied *Arid1* genotypes (An H&E stained section of each of the tumors produced was reviewed at the same sample size as the tumorigenesis experiments in **a–d** but representative photomicrographs were procured secondarily; each magnification bar is 10 μm). **g** Differential expression of SAT, **h** MAT, and **i** SST pattern genes comparing *hSS2* tumors with *wildtype* versus homozygous floxed *Arid1a* genotypes. **j** Expression heatmap of bulk RNA-seq for SST pattern genes and **k** BAF component genes among *hSS2* tumors with either *Arid1a* homozygous floxed, *Arid1b* homozygous floxed, or *wildtype* for both. Source data are provided as a Source Data file.

gel electrophoresis and transferred to PVDF membranes. Membranes were blocked with 5% milk TBST and incubated with primary antibodies overnight. After four TBST washes, blots were incubated with HRP-conjugated species-specific secondary antibodies, washed four times with TBST and developed with SuperSignal™ West Dura Extended Duration Substrate (Thermo).

### RNAseq

Fresh frozen tumor tissue was disrupted using a Tissue-Tearor (BioSpec) in 1 ml of TRIzol™ reagent (ThermoFisher). 200 μL isopropanol was added to the sample, followed by brief vortexing, and centrifugation. Supernatants were transferred into fresh tubes, to which 1 volume of Ethanol was added. Further RNA cleanup was performed using Direct-zol™ RNA Miniprep kit (Zymo), starting at step 2 of the Zymo protocol. RNA-seq libraries were prepared with the NEBNext Ultra II Directional RNA Library Prep with rRNA Depletion Kit, and sequenced on a NovaSeq using the 2 x 150 bp protocol (Illumina) for approximately 25-30 million reads per sample. Read alignment was accomplished with the STAR (2.7.11) against the mm10 version of the mouse genome[52]. Count matrices were generated using featureCounts (version 1.6.3), while differential expression analysis was performed with DESeq2 version 3.11[53]. Genes displaying a false discovery rate (FDR) below 0.05 were considered statistically significant. For the purpose of visual representation, expression levels were transformed to regularized logarithm (rlog) counts. Principal Component Analysis (PCA) plots were derived from the first two principal components based on the rlog-transformed data of the selected gene set. Additionally, heatmaps reflecting this gene set were created through unsupervised hierarchical clustering, utilizing sample Euclidean distances.

### Chromatin Immunoprecipitation (ChIP)

ChIP-seq was performed according to our previously published protocols[7]. Snap frozen tumor specimens were pulverized as previously described. Tumor pellets were crosslinked with 1% formaldehyde and quenched with 0.125 M of glycine. For double crosslinking (on samples undergoing ChIP with DPF2, PBRM1, and control samples of for SS18::SSX and ARID1A), pulverized tumors were thawed in HBSS and cross-linked with fresh 20 mM dimethyl pimelimidate (dissolved in 0.2 M Triethanolamine pH 8.2) prior to formaldehyde crosslinking. After three washes in cold PBS, nuclei were incubated in mild lysis buffer (10 mM Tris-HCl pH8.5, 10 mM NaCl, 0.5% NP-40) containing protease inhibitor (PI, Sigma), followed by wash buffer (10 mmol/L Tris-HCl pH8.5, 200 mM NaCl, 1 mM EDTA and 1% SDS with PI), and strong lysis solution step (50 mM Tris-HCl pH8.0, 10 mM EDTA and 1% SDS with PI) immediately followed by dilution 1:10 with buffer (16.7 mM Tris-HCl pH8.1, 16.7 mM NaCl, 1.2 mM EDTA, 1.1% Triton X-100, 0.01% SDS with PI). Samples were then sonicated with an EpiShear Probe Sonicator (Active Motif). 5 μg of primary antibody was then added to the sonicated chromatin for incubation at 4 °C overnight. The next day, 100 μL of washed Dynabead slurry was incubated with the IPed chromatin for 4.5 hours. Following subsequent wash and reverse crosslinking steps, DNA was purified with a DNA clean & concentrator kit (Zymo). Sequencing libraries were prepared with the NEBNext Ultra II DNA Library Prep Kit, and sequenced with a NovaSeq sequencing system using the 2 x 150 bp protocol (Illumina) for approximately 25-30 million reads per sample.

Native ChIP-seq was performed according to previously published protocols[54]. Sample were lysed and digested by MNase. Immunoprecipitation was performed using validated antibodies against H3K4me1, H3K4me3, H3K27ac, H3K27me3, H3K36me3 and H2AK119ub. DNA fragments were stripped from histones, purified, and subjected to sequencing.

Reads were aligned to mm10 mouse genome version using Novoalign (Version 3.00) for paired-end reads. Peaks were called from each of the aligned bam files against input reads using MACS2[55], (version 2.2.9) with the parameters: callpeak -B --SPMR --qvalue = 1e-3 --mfold 15 100. ChIP input was used as the background for MACS2. MACS was used to produce normalized bedgraphs, which were subsequently converted to bigWig files. Peaks were filtered to remove peaks that are in blacklist, including ENCODE blacklisted regions[56]. Duplicate reads were removed using samtools rmdup for all downstream analyses[57]. Merged bigWig enrichment files for each condition with multiple replicas were generated in an average manner followed by normalization of read depth.

Heatmaps and profile plots that illustrate the scores corresponding to genomic regions were produced using the plotHeatmap function in deepTools (version 3.5.6)[58]. This followed the computation of scores for each genomic region, a process which was executed using the computeMatrix function.

Correlation analyses of various ChIP-Seq datasets were conducted using the multiBigwigSummary function from the deepTools suite, focusing specifically on Brg1 binding sites. The Pearson correlation coefficient was employed to quantify the strength and direction of the relationship between the datasets.

### HiChIP

*hSS2* mouse tumors were harvested and digested to single cell suspension with the Miltenyi mouse tumor dissociation kit (Miltenyi Biotec). Approximately 5-10 million cells were crosslinked with 1% of formaldehyde, quenched with 0.125 M of glycine, and lysed with Hi-C lysis buffer (10 mmol/L Tris-HCl pH 7.5, 10 mmol/L NaCl, 0.2% NP40) supplemented with protease inhibitors. Cross-linked chromatin was digested by MboI restriction enzyme. The digested overhangs were then filled with dCTP, dGTP, dTTP, and biotin-labeled dATP, followed by ligation with T4 DNA ligase. Nuclei were then resuspended in nuclear lysis buffer (50 mmol/L Tris-HCl pH7.5, 10 mmol/L EDTA, 1% SDS) supplemented with protease inhibitors and sonicated with Qsonica (Q800). The sonicated chromatin was then diluted with ChIP dilution buffer (16.7 mM Tris-HCl pH 7.5, 1.2 mM EDTA, 1.1% Triton X-100, 167 mM NaCl, 0.01% SDS) prior to DNA fragment capture with H3K27ac antibody. Streptavidin C1 magnetic beads were applied to capture ligated DNA fragments, and HiChIP libraries were prepared using Illumina Tagment DNA Enzyme and Buffer Kit. Sequencing was performed with the NovaSeq sequencing system using the 2 x 150 bp protocol (Illumina).

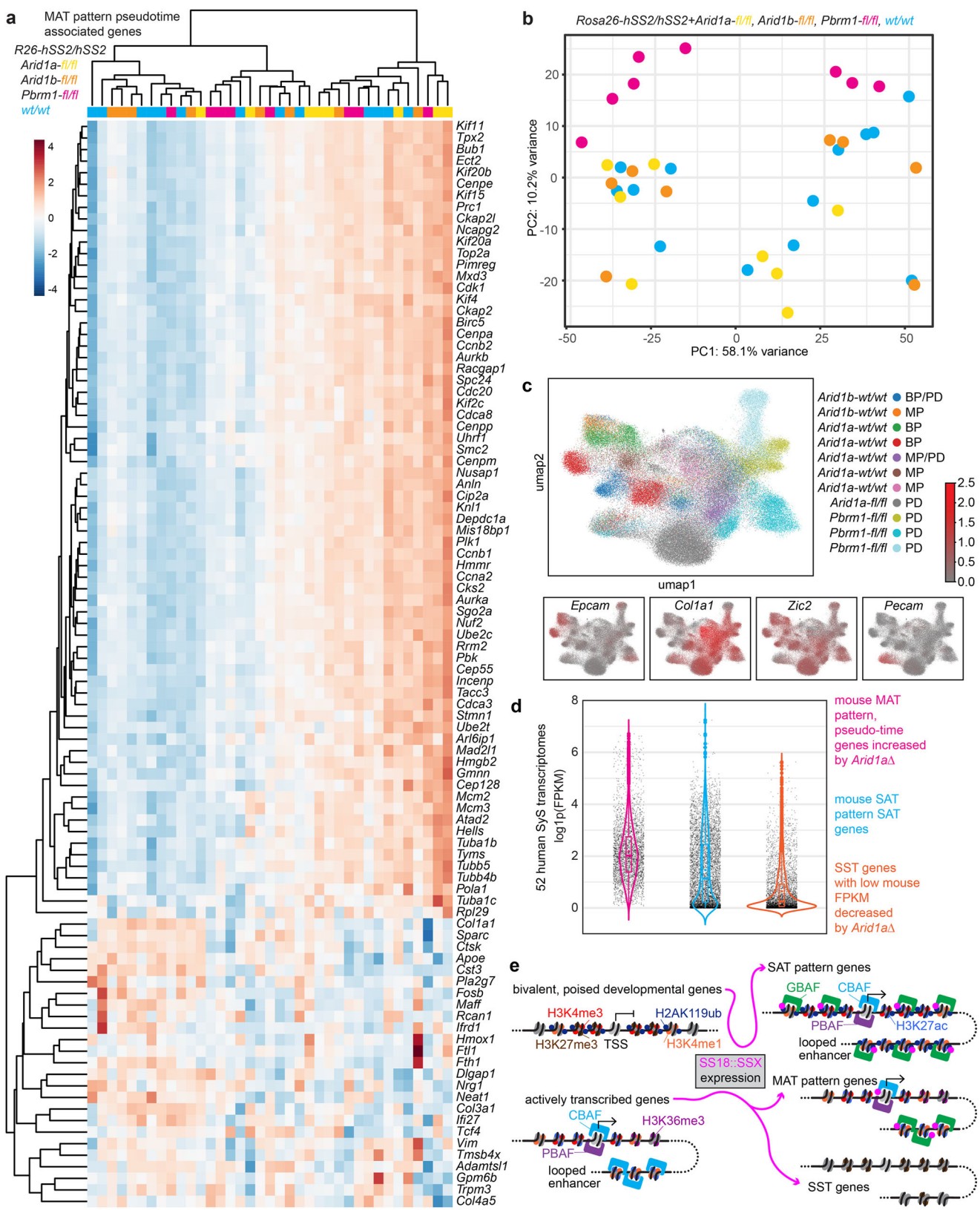

For alignment, the HiChIP sequencing data were mapped to the mouse mm10 reference genome utilizing the HiC-Pro pipeline[59]. Chromatin loops were determined using the Hichipper pipeline[60] and those with an FDR value of less than 0.05 were selected as high-confidence loops for further investigation.

Promoter regions and distal loop anchors, each spanning a 4 kb length, were pooled to define the target genomic intervals. For every target, we calculated the cumulative score (which includes both fusion and H2AK119ub markers) of the loop-associated distal enhancers and promoter region.

## Single cell RNA-seq

*hSS2* mouse tumors were harvested at morbidity. Half of the tissue was digested (as described above) using the Miltenyi mouse tumor

**Fig. 8 | Unlike Arid1a disruption, Pbrm1 disruption alters synovial sarcoma transcriptional features. a** Expression heatmap for non-SAT pattern genes identified as associated with pseudo-time trajectories, tested by bulk RNA-seq of *hSS2* tumors with varied *Arid1a*, *Arid1b*, and *Pbrm1* genotypes. **b** PCA plot of whole transcriptomes for *hSS2* tumors with varied *Arid1a*, *Arid1b*, and *Pbrm1* genotypes. **c** Xenium In Situ single cell transcriptomic UMAPs, indicating specific clusters that are epithelial or mesenchymal neoplastic populations of cells, versus endothelial cells, and how much each of the unique sample sources contribute to the varied clusters. **d** Expression of three gene sets among 52 human SyS transcriptomes (biological replicates, individual tumors arising in individual patients) from our human dataset, showing conserved expression of CBAF-loss associated gene activations and repressions (boxes indicate 25th to 75th percentile range and median;

diamond indicates mean; whiskers indicate minima and maxima; each two-way comparison had $p < 2.22 \times 10^{-16}$ on Wilcoxon rank sum tests. Sample sizes of 52 independent human tumor samples for MAT genes, $n = 68$, SAT genes, $n = 113$; SST genes, $n = 332$). **e** Working model of transcriptional regulation by SS18::SSX, wherein genes with promoters and enhancers that are poised, exhibiting H3K27me3/H3K4me3 bivalency in development, are shifted towards expression by the presence of SS18::SSX in cells, which promotes H3K4me3 monovalency following the redistribution of GBAF complexes containing the fusion. Other genes that are expressed at baseline, are silenced during sarcomagenesis from expression of the fusion and subsequent CBAF desertion from its typical distribution across the genome-wide chromatin of normal cells. Other genes maintain expression and retain CBAF enrichment, but only narrowly near TSSs, where PBAF is also present.

dissociation kit (Miltenyi Biotec). The remaining tissue from each sample was embedded in OCT for H&E staining in order to classify the corresponding tumor phenotype. After dissociation the cell suspension was filtered through a 40 μm strainer and treated with red blood cell lysis solution (Miltenyi Biotec). Lastly, cells were resuspended in PBS + 0.04% BSA at a concentration of ~1,000 cells/μL for single-cell sequencing. 10,000 cells per sample were targeted. scRNA-seq libraries were prepared using Chromium Next GEM Single Cell 3′ Reagent Kits v3.1 according to the User Guide (CG000204), followed by amplification with paired-end dual-indexing (28 cycles Read 1, 10 cycles i7, 10 cycles i5, 90 cycles Read 2). The cellranger count command was employed for demultiplexing, barcode processing, and single-cell 3′ gene counting, resulting in a filtered matrix of gene expression counts for individual cells.

Quality control, normalization, and downstream analysis were performed using the Seurat package (v5.0.3) in R. We plotted the distribution of mitochondrial percentages and nFeature_RNA values to guide threshold selection, and then tested different cutoffs to remove poor-quality and artifactual events. Very stringent cutoffs (5–10%) tended to exclude an excessive number of cells, including metabolically active populations such as malignant and stromal cells. Key results remained stable across thresholds of 20–30%. Because our samples were derived from fresh solid tumors, enzymatic dissociation likely led to elevated mitochondrial transcript fractions by preferentially depleting cytosolic RNA. We therefore applied a relaxed global filter of %mt ≤ 25%, which removed overtly damaged cells while retaining true tumor and microenvironmental populations. To further avoid including low-quality cells, cells with fewer than 750 detected genes were removed, and DoubletFinder was applied to identify and discard putative doublets. Data were then normalized using the SCTransform function[61]. Cluster markers were identified using the FindMarkers function, with a minimum log-fold change of 1, pct.1 of 0.3 (expression of a gene in more than 30% of cells within the target cluster) and an adjusted p-value threshold of 0.01.

For pseudotime analysis, processed scRNA-seq data were imported into Monocle 3 (version 1.3.4) (http://cole-trapnell-lab.github.io/monocle-release/monocle3), and converted to cell data set format with the SeuratWrappers Package (version 0.3.2).

## Xenium in situ

Seven *hSS2* mouse tumors (with histologically different morphology), one *hSS2; Ariad1a*^fl/fl and three *hSS2; Pbrm1*^fl/fl tumors were selected for Xenium In Situ analysis on FFPE tumor sections. 348 genes were designed for cell type identification (248 of Xenium pre-designed mouse Brain Panel and 100 customer designed SAT genes[31]). Five-μm-thickness FFPE sections were placed onto a Xenium slide, followed by deparaffinization and permeabilization for mRNA accessibility. Probe hybridization, Ligation and amplification were performed according to the Xenium In Situ GeneExpression User Guide (CG000582) prior to

processing within the the Xenium Analyzer Instrumentation pipeline. Data generated by this pipeline was further analyzed downstream. Cell segmentation processing was performed using the cell segmentation user guide (CG000749), and post-Xenium tissue H&E staining was performed according to user guide CG000613.

Data integration and normalization steps were carried out in python package Scanpy[62] to correct for technical variance and to normalize expression counts. Spatial expression data were merged across different sections, with normalization to align gene expression levels.

## Single cell clustering and UMAP visualization

Cell clustering was performed using a shared nearest neighbor (SNN) modularity optimization algorithm[63]. Louvain community detection was applied to the PCA-reduced data (30 principal components) to identify distinct transcriptional clusters within the tissue. For spatial data visualization, UMAP (Uniform Manifold Approximation and Projection) was applied to generate two-dimensional embeddings that reflect the spatial gene expression patterns.

## Access to human databases

Human genomics data from SySs was accessed via cBioPortal using the AACR Project Genie data and the Memorial Sloan-Kettering combined cohorts[64–66]. The single gene dependencies for human SyS cell lines were accessed on the DepMap web interface at https://depmap.org/portal using 25Q2 public data[46].

## Statistics and reproducibility

For the comparison of means between two independent groups, we utilized the two-tailed Student's t-test, as facilitated by GraphPad Prism software (version 9.0). We established statistical significance at p-values of less than 0.05 or 0.01, which are specified accordingly in the legends accompanying each figure. All data are depicted as the mean ± standard deviation. The determination of sample sizes was guided by the variability observed in preliminary experiments. Notably, the numbers **n**, referenced in the figure legends correspond to the count of biological replicates rather than repeated measures of the same specimen. We have employed additional statistical methods as necessary, with the specifics of these analyses noted within the legends of each figure. Furthermore, statistical procedures pertinent to genomic data have been detailed in the respective analysis subsections under the Methods section.

Statistical significance in RNAseq data was determined using the Wald test for pairwise comparisons, and P-values were adjusted for multiple testing using the Benjamini–Hochberg false discovery rate (FDR) method. Genes with an adjusted $P < 0.05$ (FDR < 0.05) were considered significantly differentially expressed. Non-parametric Kruskal-Wallis H tests were applied to assess the statistical significance of independent groups where the assumption of normality was not met. Dunn's test was used for post-hoc analysis after a Kruskal-

Wallis H test indicated a significant result. Bonferroni correction was used to adjust the significance levels.

All key experiments were independently repeated 3 times with consistent results. For genomic, imaging, and animal studies, biological replicates were performed using independent samples.

## Reporting summary

Further information on research design is available in the Nature Portfolio Reporting Summary linked to this article.

## Data availability

All data supporting the findings of this study are available within the paper and its Supplementary Information. All sequencing data generated in this study have been deposited in the NCBI Gene Expression Omnibus (GEO) under the following accession codes:

GSE269770 (ChIP-Seq and HiChIP) https://www.ncbi.nlm.nih.gov/geo/query/acc.cgi?acc=GSE269770;

GSE269772 (RNA-Seq) https://www.ncbi.nlm.nih.gov/geo/query/acc.cgi?acc=GSE269772;

GSE269773 (scRNA-Seq) https://www.ncbi.nlm.nih.gov/geo/query/acc.cgi?acc=GSE269773

Processed signal files (BigWig format) are publicly accessible through GEO. All other processed datasets and source data supporting the figures are provided in the Supplementary Information. No restrictions apply to data availability. Source data are provided with this paper.

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

## Acknowledgements

Work was supported by the L.B. and Olive S. Young Presidential Chair for Cancer Research and the University of Utah Department of Orthopaedics (to K.B.J), the Howard Hughes Medical Institute (to B.R.C.), the Huntsman Cancer Foundation and R01CA201396, U54CA231652 and 2P30CA042014-31 from the National Cancer Institute (NIH) (to K.B.J. and B.R.C). T.O.N., M.H., and T.M.U. were supported by the Terry Fox Research Institute (Program Project grant 1155) and Canadian Institutes for Health Research (PT025845). We thank Brian Dalley and the High-Throughput Genomics Core at Huntsman Cancer Institute for sequencing support and Tim Parnell and the Bioinformatics Core at Huntsman Cancer Institute for sequencing alignments.

## Author contributions

B.R.C. and K.B.J. conceived of the overall experiments in consultation with T.O.N., M.H., and T.M.U. Experiments were performed by J.L., K.S.F., L.M., X.G., M.N., L.A.H., Y.G., and G.D. X.Z. consulted on techniques. L.L., J.L., Z.F.W. performed bioinformatics analysis with consultation from M.H., X.Z. and K.B.J. K.B.J., B.R.C., T.M.U., M.H., and T.O.N. procured funding and leadership over the projects. K.B.J. wrote the manuscript with assistance from J.L., L.L., and L.C. All authors reviewed, edited, and approved the final manuscript.

## Competing interests

The authors declare no competing interests.
