## [Transparent Peer Review file · Nature Communications]

Synovial sarcoma reprograms transcription by GBAF activation of polycomb targets and loss of CBAF enhancers

Corresponding Author: Dr Kevin Jones

Version 0:

Reviewer comments:

Reviewer #1

(Remarks to the Author)
Comments to the Authors:

In Li et al, authors use a murine model system to explore synovial sarcomagenesis and function of SS18::SSX fusion protein. The authors probe the genomic profiles of various histone modifications and BAF complex subunits in these tumors and use scRNA-Seq to map transcriptional profiles of various cells within the tumors. Through these experiments, they seem to conclude that a specific complex (GBAF, which is more frequently called “ncBAF” in the literature) but not the related CBAF (which should be “cBAF” per other labs) or PBAF complexes, are recruited to genomic loci marked by H2AK119ub. This has all been shown before, because this mechanism demonstrated in previous works (PMID: 38177667, PMID: 32747783) is based on SSX and anything SSX is bound to (which has been shown to be ncBAF and a disrupted/residual CBAFF, not PBAF). These authors interpret their data as to mean that CBAF and PBAF complexes occupy narrow sites at TSS of active genes and “avoid” H2AK119ub through an unknown mechanism. This reviewer does not find their claims substantiated by the data presented. Further, these aspects of the study are not novel beyond what is published.

The authors use reverse genetic experiments to demonstrate that PBAF-specific PBRM1 loss blocks synovial sarcomagenesis while ARID1A or ARID1B loss enhances sarcomagenesis. Overall, the mechanistic basis of the interactions between these different complexes in SyS is lacking throughout the manuscript, which is often quite descriptive in nature. Genomics experiments lack a control cell population, not expressing SS18::SSX, making it difficult to assess the changes associated with the expression of fusion protein relative to a “normal” (or non-fusion-containing) cell population. In genetic experiments, the CBAF-specific ARID1A or ARID1B subunits (which are somewhat-redundant in forming CBAF) were depleted individually and not together, failing to completely disrupt CBAF complexes in these SyS cells, making certain results presented here uninterpretable.

With the absence of an overall mechanistic insight into the functions of BAF complexes in sarcomagenesis and experimental issues including of lack of control cell lines, or complete disruption of CBAF complexes and others as outlines below, this manuscript does not warrant publication in Nature Communications.

Major specific comments:

1. Authors utilized murine tumor model system to study SyS, as they claim human patient-derived cell lines are aplastic. As a result of the experimental design, the analyses presented in figure 1 and 2 completely lack a control “normal” population of cells that do not express the SS18::SSX fusion protein. In the absence of control “normal” cells, it is difficult to attribute any chromatin occupancy profiles of histone modifications and BAF complexes to the SS18::SSX fusion protein. This is a major limitation of the work.

2. Authors performed ChIP-Seq experiments on mouse tumors, which represent a heterogenous cell population of cells from the data presented – a fraction of cells express SS18::SSX and another fraction does not express SS18::SSX. The genomic occupancy profiles of BAF complex subunits (DPF2, BRD9, PBRM1 and SMARCA4) in these experiments represent a

population average. Therefore, the sharp peaks of CBAF and PBAF, and broader peaks of CBAF and GBAF might actually be representative of distinct nontumor cells within the tumors, and it is not possible to show which have and which do not have SS18::SSX expression. Given the choice of tumor model system, rather than a homogenous cell line system, authors would need to perform single-cell CUT&TAG or another single-cell technique to address this issue.

3. The results presented in figure 1 – SS18::SSX1 fusion protein localized with H2AK119ub1 in cells – are already well known from previous studies that used patient-derived cancer cell lines and do not add any novel mechanistic insight into the function of SS18::SSX. See PMID: 38177667, PMID: 32747783.

4. Authors probe occupancies of CBAF, PBAF and GBAF specific subunits, DPF2, PBRM1 and BRD9, respectively as a proxy for localization of each complexes. However, it remains unclear as to what fraction of DPF2 is free subunit and what fraction incorporates into full-assembled complexes in the presence of SS18::SSX fusion protein. DPF2 western blot is missing from the figure 2e. Further, it has been shown previously that DPF2 is not a component of CBAF complexes in the absence of SMARCB1 (see PMID: 29861296, PMID: 28945250, and PMID: 27941797)

5. Along these lines, what fraction of CBAF and GBAF complexes incorporate SS18::SSX fusion proteins? Is the DPF2 that is pulled-down in ChIP-Seq experiments assembled into CBAF complex or are they 'free' DPF2 simply interacting with nucleosome modifications at active promoters (DPF2 contains double PHD domains)? Clarifying this point is essential to make any claim about the "CBAF complex occupancy" especially since ARID1A/ARID1B ChIP-Seq seemingly did not work in their experiments. Again, homogenous populations of cells are also needed to derive conclusions here.

6. Authors should perform ChIP-reChIP experiments with SS18::SSX1 antibody followed by CBAF- and GBAF- specific antibodies to determine the occupancies of these complexes containing SS18::SSX1 fusion protein. Similarly, they would need to perform ChIP-Seq experiments with a SMARCA4 antibody, followed by CBAF/GBAF/PNBAF-specific subunit antibodies to assess the occupancies of CBAF complexes as opposed to free subunits.

7. Authors would be expected to validate their findings regarding occupancies of CBAF, GBAF and PBAF complexes in mouse tumors in synovial sarcoma lines. All the experiments in figure 1 and 2 are performed at equilibrium, when the tumor state has already been achieved. Therefore, this model is as much cellularly aplastic as patient-derived cell lines.

8. Authors claim that DPF2 (and by proxy, CBAF) "avoids H2AK119ub-marked sites" (in abstract), "often accompanying SS18::SSX" (line 205). What is the mechanism by which CBAF containing SS18::SSX, which biochemically interacts with H2AK119ub, avoids H2AK119ub? The finding that SS18::SSX (not wild-type or normal) CBAF would avoid H2AK119ub, while unsubstantiated by any biochemical evidence presented to date, is the only novel potential finding in figures 1 and 2; authors would need to provide a mechanistic basis for this avoidance that is not just a technical artefact of their experiments.

9. "...CBAF at TSS **correlates** with gaps in the H2AK119ub and H3K4me3 distribution..." (line 206) is the only basis for the main conclusion that CBAF avoids H2AK119ub in SyS. However, it appears that CBAF at TSS also correlates with gaps in SMARCC1, which is a subunit in the same complex, at all sites (supplementary Fig 2j at every single example presented). Does this mean CBAF also avoids SMARCC1 in SyS? These data are very weak and correlative and the degree of conclusions derived from these are not substantiated by the data presented.

Perhaps the authors should consider that these conclusions from genomics experiments alone, in absence of any biochemical experiments, are perhaps being over-interpreted.

10. Authors note that DPF2 (and by proxy CBAF) distributes heavily to promoter regions in SyS, as opposed to murine ES cells. What is the biochemical mechanism by which expression of SS18::SSX leads to such redistribution of DPF2 (or CBAF)? Especially when DPF2 is not even present in CBAF complexes as shown before (see reference above).

11. Authors claim that CBAF is depleted in SyS (Fig 2e), which was previously reported in another publication from this lab, where knockdown of SSX increased appearance of CBAF complex subunits on glycerol gradient. However, CBAF containing SS18::SSX has been demonstrated to interact strongly with chromatin marked with H2AK119ub even at high salt concentrations in structural and biochemical studies by the Liu and Kadoch labs. Therefore, the apparent loss of CBAF in this study and the other study is very likely an artefact of protein extraction method.

Authors should carefully fractionate chromatin-bound and unbound fractions of nuclear complexes, followed by nuclease treatment to demonstrate if SS18::SSX causes depletion of CBAF complexes. A simple whole cell extract preparation using laemmli buffer might also resolve this issue, as Laemmli buffer efficiently extracts all proteins from chromatin fraction unlike salt.

12. What do the RNA-Seq profiles of human SyS cell lines look like compared to the scRNA-Seq here? Why not do scRNA-Seq in patient derived tumor samples? Isn't that the point of these experiments?

13. Authors claim that SS18::SSX fusion protein functions predominantly through recruitment of GBAF complex to H2AK119ub, rather than CBAF. However, in rest of manuscript, they delete CBAF- and PBAF-specific subunits but not any GBAF subunit. If their claim is correct, SS18::SSX expression should fail to form SyS in mouse model. Authors would need to generate this mouse model to support their (very strong) claim.

14. Fig 4: Please perform ChIP-Seq for all BAF subunits in smarcb1f/f and wt/wt cells, not just SMARCA4. Why does

SMARCA4 reduce substantially (~2-3x), as loss of SMARCB1 shouldn't have any effect of GBAF and CBAF, which are enriched at these sites?

15. PBAF complexes localize to the promoters of active genes and sustain their transcription in all cells, including SyS. Unsurprisingly, PBRM1 knockout in their SyS model altered the typical SyS phenotype. What is the relevance of these findings in SyS?

16. What is the mechanism by which ARID1A or ARID1B disruptions enhance SyS tumorigenesis (Note that CBAF is not actually disrupted in either of these cases because of paralog redundancy; see below)? ARID1A or ARID1B loss enhances SyS sarcomagenesis. However, LOF mutations in these proteins are never found in early- or late- stage SyS tumors from human patient, which would be a prediction from this study?

17. ARID1A and ARID1B are two paralogs, that both each assemble into CBAF complexes mutually exclusively. Likewise, loss of one can be compensated by the other paralog as has been shown (Helming et al., Nat Med 2013). Authors only deleted one of the two paralogs in all the experiments performed in figure 6, never both. Therefore, CBAF is not necessarily disrupted in these cells as the remaining paralog compensates for the loss of one. In the absence of ARID1A/ARID1B double knockout, the results presented in figure 6 and 7 are un-interpretable.

18. Authors should test these genetic findings presented in Fig 4-7 in patient-derived SyS cell lines. Does loss of ARID1A and ARID1B enhance proliferation of SyS cell lines? Does PBRM1 deletion in SyS cell lines change their phenotype at all?

Reviewer #2

(Remarks to the Author)

Li and colleagues addressed the complex issue of epigenomic dysregulation mechanisms in synovial sarcoma (SyS). The SS18::SSX oncoprotein substitutes SS18 in CBAF and GBAF complexes, leading to preferential CBAF reduction as previously shown by the same research group (Li et al., Cancer Discov, 2021) and genome wide altered location of GBAF complexes. The SSX domain of the fusion plays a role in GBAF redistribution, indeed the SSX domain directly binds in a peculiar way to nucleosomes which are marked with H2AK119ub1 (Tong et al., Nat Struct Mol Biol, 2024) for silencing.

In this paper, the Authors show that SS18::SSX targets promoters in two patterns, narrow and broad, with broad promoter binding for sarcomagenesis activated transcription (SAT) genes and narrow binding for genes that are maintained active before and after transformation (MAT genes). SS18::SSX expression redistributes GBAF complexes broadly to promoters and distal enhancers marked by H2AK119ub as evinced by the strong coincidence of H2AK119ub and the fusion in histone mark and fusion ChIP-seq enrichment profiles. Binding of GBAF complex causes H3K27me3 loss and makes transcriptionally active those developmental loci, previously marked for silencing, leading to inappropriate activation of developmental genes that induce SyS onset and progression. In addition CBAF containing SS18::SSX avoids H2AK119ub-marked sites, and instead distributes with PBAF narrowly to transcription start sites as highlighted by the strong correlations found between PBAF and CBAF distributions in these sites.

The Authors frequently quote and refer to two paired papers from their own group not yet published but only available online as preprints on bioRxiv. In one of these papers, namely in "Synovial Sarcoma Chromatin Dynamics Reveal a Continuum in SS18::SSX Reprogramming" by Hofvander et al, they analyze the mechanisms of action of the fusion transcript SS18::SSX in human synovial sarcoma. The present work further analyzes transcriptional reprogramming by exploiting synovial sarcomagenesis in mouse models. The Authors take advantage of different SyS conditional mouse models on hSS2 (SS18::SSX2) or hSS1 (SS18::SSX1) background with genetic disruption of Smarcb1 (PBAF and CBAF component), or Pbrm1 (PBAF component), or Arid1a and Arid1b (both CBAF components), to investigate how each of them impacts on CBAF, GBAF and PBAF destiny and activity, and affects tumorigenesis, adding insights into the role of different components of BAF chromatin remodeling complexes. Loss of these elements speeded up tumorigenesis in hSS2 mice, loss of Smarcb1 and Pbrm1 changed tumor morphology while loss of Arid1a and b did not. They conclude that synovial sarcomagenesis requires both SS18::SSX action on transcription reprogramming through GBAF-mediated reactivation to polycomb-targeted developmental genes, and reduced presence and altered localization of CBAF. They further suggest that the focal localization of CBAF and PBAF at the TSS requires SMARCB1.

Overall results are convincing, significant and relevant to the topic. The work is original and further develops established literature. The methodology is largely correct and meets the accepted standards in the field. Details are fully provided in the methods for the work to be replicated.

However, the following points are identified and need to be addressed for revision before publication in Nat Commun.

Major points:

1) The Authors cite to two paired papers from their own group not yet published but only available online as preprints on bioRxiv. Indeed, the reading of these two articles is of help to the understanding of the present paper and they should be more extensively recapitulated in the paper. The authors should better describe how and in which model the three gene categories (SAT, MET and SST) were identified.

2) The hSS1;Pbrm1-fl/fl mice behave differently from hSS2;Pbrm1-fl/fl mice as regard tumorigenesis. The Authors conclude that "this apparent paradox suggests that Pbrm1 silencing similarly blunted synovial sarcomagenesis programs in both SS18::SSX backgrounds. This blunting of SyS programs tipped the faster growing hSS2 tumors into an alternate tumorigenesis program, but merely slowed hSS1 tumor development, causing selection for alleles that escaped

recombination". Any role of a differential immune signature between the two backgrounds?

- 3) The Authors state that: "Having firmly established the expression of SAT gene signatures in human SyS in our related manuscripts^{29, 31}, we next sought to confirm that these CBAF-loss related signatures of gene expression and repression are also conserved between species with SS18::SSX-driven malignancy. The SST genes that were silenced more profoundly by Arid1a genetic disruption were found to be very lowly expressed in 52 human SyS transcriptomes. Inversely, the pseudo-time trajectory associated MAT pattern genes that were further upregulated by Arid1a silencing were highly expressed in the same human SyS tumors". Are these "MAT pattern genes" involved in SyS tumor progression?
- 3) "As future therapeutic efforts variably target each of these mechanisms (gain of GBAF function at polycomb targets or loss of CBAF function), the gene sets we have identified as associated with each constitute vital datasets to interrogate mechanisms and efficacies of therapeutics". This issue should be expanded mentioning possible drugs targeting H3K4me3, or H2AK119ub, or BAF complexes, and eventually related gene expression data if available.

Minor points:

- 1) In Figure 1, b,c,d,e, drawings need to be arranged in a row or grouped in order to be more readable
- 2) In the results, with reference to Figure 1, h-m, please comment on H3K36me3
- 3) In Figure 3, a, please check for the indications of the five source tissue types
- 4) In Figure 6, f, please check the nomenclature of each picture
- 5) In methods "Single cell RNA-seq" : "Cells were filtered based on quality metrics, including removal of cells with mitochondrial features surpassing 25%, or total features below 750." Please explain why using the parameter of 25% and not the more selective 10% or 5%?
- 6) Please make sure that References are reported uniformly and according to the journal requirements.

Reviewer #3

(Remarks to the Author)

The manuscript by Li and Li et al. entitled "Synovial sarcoma reprograms transcription by GBAF activation of polycomb targets and loss of CBAF enhancers" uses a previously described transgenic mouse model for the SS18::SSX oncogene – the dominant driver of synovial sarcoma (SyS) to study how this fusion oncoprotein promotes a marked redistribution of non-canonical GBAF complexes to promoters and distal enhancers marked by H2AK119ub, which causes so-called "developmental loci" to lose H3K27me3 and consequently become active. Using a series of ChIP-seq and HiChIP experiments, the authors demonstrate that canonical BAF complexes containing SS18::SSX abandons its typical binding sites and redistributes close to PBAF at TSSs. For this phenotype, the CBAF-specific components Arid1a or Arid1b appear to be not relevant as the SyS phenotype in mice is largely unaltered. However, the PBAF- and CBAF-specific components Smarcb1 or Pbrm1 (PBAF-specific) has an impact on SyS formation with accelerated tumorigenesis driven by SS18::SSX. In sum, this is an elegant study combining state-of-the-art omics technologies to uncover how SS18::SSX reprograms transcription through GBAF redistribution in transgenic SyS mouse model. The paper is well-written, and the conclusions are largely supported by the data. There are only a few concerns/aspects that should be addressed before publication:

Major comments:

1. The single-cell and spatial transcriptomic data shown in Figure 3 are very interesting. Yet, it would be advisable to confirm the spatial expression of some of the proposed markers at the protein level by IHC in tumor tissue.
2. While this study certainly has the groundwork for more translational studies, the current manuscript could benefit from some analyses to which the authors already allude to in lines 390-392: "This observation fits a model for heterogeneity in human SyS, where tumors that have more genomic copy number variation also have more retained bivalency, as shown in our human datasets." Do SyS tumors also show in some cases alterations of SMARCB1 and/or PBRM1 that could translate into variable clinical phenotypes? Can the authors elaborate on which CNVs may contribute specifically to the more retained bivalency? Are the transcriptional features and signatures shown in Fig. 7 of prognostic relevance?
3. Recent data in epithelioid sarcoma – a SMARCB1 deficient cancer – has shown that the residual SWI/SNF complex has an altered but still active effect on the transcriptional and epigenetic reprogramming of these tumor cells that may constitute a specific vulnerability of BRG1 inhibition (PMID 39834137). It would be very interesting to explore whether SS18::SSX confers a similar vulnerability to SyS cells/tumors.

Minor comments

1. Line 32 abstract: should read SS18::SSX (not SS18:SSX)
2. Line 50: "...expression of one of these fusions..." >> Which one(s)?

Version 1:

Reviewer comments:

Reviewer #1

(Remarks to the Author)

Thank you for addressing reviewer comment.

Reviewer #2

(Remarks to the Author)

The authors addressed most of my previous concerns. The results and figures are more accurate and the conclusions are now clear.

Reviewer #3

(Remarks to the Author)

The authors sufficiently addressed all my concerns.

We would like to thank the three reviewers for their helpful comments. Our revisions to the manuscript in response to the noted critiques have improved the rigor of our science and the clarity of our message communication. We have also enhanced the registration of our findings in the mouse more deeply with human synovial sarcoma data, to address comments from each of the Reviewers. Below, we have added point-by-point responses (in blue text) to each comment written in the critiques.

Reviewer's Comments:

Reviewer #1 (Remarks to the Author): Expert in synovial sarcoma, gene fusion, BAF complexes, chromatin remodeling, epigenetics and scRNA-seq

Comments to the Authors:

In Li et al, authors use a murine model system to explore synovial sarcomagenesis and function of SS18::SSX fusion protein. The authors probe the genomic profiles of various histone modifications and BAF complex subunits in these tumors and use scRNA-Seq to map transcriptional profiles of various cells within the tumors. Through these experiments, they seem to conclude that a specific complex (GBAF, which is more frequently called "ncBAF" in the literature) but not the related CBAF (which should be "cBAF" per other labs) or PBAF complexes, are recruited to genomic loci marked by H2AK119ub.

Nomenclature of SWI/SNF complexes and components has been controversial for 3 decades. With regard to GBAF versus ncBAF, we have simply respected the naming convention of the first publication that correctly identified this complex (Aktan Alpsoy and Emily Dykhuizen in March of 2018 in the *Journal of Biological Chemistry* named it GBAF. PMID: 29374058). We explain the acronyms used for these complexes in our paper—as well as the alternative acronyms used in the literature, which are now all added—at our first mention in the text (page 3, Introduction paragraph 3).

Another important point that we have more thoroughly addressed in our revised Discussion (page 11, Discussion paragraph 3) was brought up by the Reviewer's comment that we had concluded that CBAF is *not recruited* to H2AK119ub. Such a conclusion is not supported by our findings. We can only appropriately conclude that CBAF complexes *do not co-distribute* with H2AK119ub *consistently* across chromatin, as GBAF complexes do. We expect that CBAF with the fusion is avidly recruited to H2AK119ub-decorated chromatin by the SSX tail, as has been shown by many others, biochemically. However, something physically or chemically hinders the sustained presence or activity of CBAF with the fusion at H2AK119ub-decorated sites, leading to the destruction or removal of many thus-recruited complexes. This is a very critical point and we thank Reviewer #1 for bringing clarity to our communication of this point. Indeed, we do not think that our genomics data in any way challenge the excellent biochemistry regarding the relationship between the SSX tail and H2AK119ub that has been done by others.

This has all been shown before, because this mechanism demonstrated in previous works (PMID: 38177667, PMID: 32747783) is based on SSX and anything SSX is bound to (which has been shown to be ncBAF and a disrupted/residual CBAFF, not PBAF).

The first of the two articles cited includes an elegant structural interrogation of the interaction between the SSX tail and the nucleosome acidic patch, especially when H2AK119 is monoubiquitinated. The second of the two articles cited is a superb biochemical evaluation of this binding interaction between

the SSX tail and H2AK119ub on the nucleosome acidic patch. We cite both of these papers, as well as a third, not mentioned by the Reviewer (PMID: 37735617), that also digs into the cellular mechanism of H2AK119ub functioning as a recruiting signal for the fusion in the cellular biology of synovial sarcoma. While the second paper cited included some mapping of the fusion oncoprotein across chromatin after exogenous expression in a human fibroblast cell line, neither cited article reports the distribution of the fusion in actual tumors resulting from fusion oncoprotein expression. We propose that there are insights to be gained from experimental platforms and paradigms beyond biochemistry alone. Our mouse models provide an *in vivo* correlate to this biology that corroborates the prior biochemistry and cell line efforts in a tumorigenesis context that also lends additional new physiobiological insights. (see page 12 Discussion paragraph 5).

These authors interpret their data as to mean that CBAF and PBAF complexes occupy narrow sites at TSS of active genes and “avoid” H2AK119ub through an unknown mechanism. This reviewer does not find their claims substantiated by the data presented. Further, these aspects of the study are not novel beyond what is published.

This reviewer judges our claims to be insufficiently substantiated which is incongruent with a judgement of their lacking novelty. It is the very clear departure from what has already been published that is novel in our results. With regard to substantiation, we will detail below why we disagree with this assessment of the data in question (with new results added in the current revision), which we hope will be convincing to Reviewer #1.

The authors use reverse genetic experiments to demonstrate that PBAF-specific PBRM1 loss blocks synovial sarcomagenesis while ARID1A or ARID1B loss enhances sarcomagenesis.

Overall, the mechanistic basis of the interactions between these different complexes in SyS is lacking throughout the manuscript, which is often quite descriptive in nature.

If by mechanistic basis, the Reviewer is discussing a biochemical mechanism, indeed our manuscript does not report biochemical mechanistic experiments, many of which have already been performed by others (such as the relationship between SSX and H2AK119ub-decorated nucleosomes, discussed above). This is not because we do not value mechanistic biochemistry, as one of our senior authors has spent over 30 years dissecting the mechanistic biology of SWI/SNF complexes. We simply value other experimental paradigms as well. The genomic profiling and developmental genetics of tumorigenesis offer unique insights not possible from biochemistry alone.

Genomics experiments lack a control cell population, not expressing SS18::SSX, making it difficult to assess the changes associated with the expression of fusion protein relative to a “normal” (or non-fusion-containing) cell population.

As explained in the original manuscript—but better emphasized now in the revision—our initial control cell population is indeed the ideal control: the *Hic1⁺ Pdgfra⁺ Lgr5⁺* fibroblastic mesenchymal progenitor cells that exhibit specific and strong origination potential for synovial sarcoma in one of the mouse models derived from expression of SS18::SSX. This control cell population is described in detail in a companion paper (Hill et al., bioRxiv, 2024

<https://www.biorxiv.org/content/10.1101/2024.05.15.594021v1>), now pending publication at *Nature Communications*. We have added an explanatory figure (new Fig. 1) summarizing the foundational findings of our two companion papers for reference in the beginning of this paper, to make it easier for

readers to understand the background on which this paper builds. We define promoters of interest through the strict comparisons of mouse tumor cells to the control cells of origin as detailed in the companion manuscript and now noted in Fig. 1. Nonetheless, for the epigenomics experiments in the first two prior figures (now Fig. 2 and 3), rather than a control cell population, the loci of each specified character are compared to loci of another specified character as control loci.

In genetic experiments, the CBAF-specific ARID1A or ARID1B subunits (which are somewhat-redundant in forming CBAF) were depleted individually and not together, failing to completely disrupt CBAF complexes in these SyS cells, making certain results presented here uninterpretable.

We agree that interpretation of ARID1A or ARID1B disruption as tantamount to complete CBAF disruption would be overstatement. We have adjusted the language in the manuscript to avoid strictly any suggestion that ARID1A or ARID1B disruption generates CBAF loss. However, perfect redundancy of ARID1A and ARID1B is not a realistic biological expectation beyond their redundant roles in biochemical assembly experiments. Importantly, the expression of neither was increased by disruption of the other in a way that suggested compensation *in vivo*. Further, as *ARID1A* is the most common single gene loss (in tumor suppressor fashion) among the genes encoding subunits for the entire family of BAF complexes—with gene loss frequency of *ARID1B* not far behind—we argue that these results are indeed interpretable. The loss of either of these majorly impactful tumor suppressors had no real impact on synovial sarcomagenesis, with regard to tumor phenotype, beyond a slight acceleration of its pace of development. We directly contrasted this result with two other gene losses that critically impacted synovial sarcomagenesis. From the standpoint of genetic developmental tumorigenesis experiments, these results are both profound and informative. They demonstrate the clear experimental result that neither *Arid1a* nor *Arid1b* is important as a single gene for synovial sarcomagenesis with a well-controlled, rigorous experimental design for understanding SyS developmental genetics. (page 12, Discussion paragraph 4)

With the absence of an overall mechanistic insight into the functions of BAF complexes in sarcomagenesis and experimental issues including of lack of control cell lines, or complete disruption of CBAF complexes and others as outlines below, this manuscript does not warrant publication in Nature Communications.

We will address the specific items considered to be weaknesses, below, but defer to the Editors whether or not the clearly novel insights gained from these rigorous developmental genetic experiments merit publication in *Nature Communications*, which is not a subspecialty biochemistry journal.

Major specific comments:

1. Authors utilized murine tumor model system to study SyS, as they claim human patient-derived cell lines are aplastic. As a result of the experimental design, the analyses presented in figure 1 and 2 completely lack a control “normal” population of cells that do not express the SS18::SSX fusion protein. In the absence of control “normal” cells, it is difficult to attribute any chromatin occupancy profiles of histone modifications and BAF complexes to the SS18::SSX fusion protein. This is a major limitation of the work.

We agree that control cells are of utmost importance to the study of synovial sarcomagenesis. We have added a more significant reference to our companion manuscript in the new Figure 1, that explains how we identified a very specific cell of origin for synovial sarcoma and used these cells as a control for fully developed tumors. We would posit that the transcriptomic changes from these experiments, which are detailed in our companion manuscript, are a major strength of our work, overall.

2. Authors performed CHIP-Seq experiments on mouse tumors, which represent a heterogenous cell population of cells from the data presented – a fraction of cells express SS18::SSX and another fraction does not express SS18::SSX. The genomic occupancy profiles of BAF complex subunits (DPF2, BRD9, PBRM1 and SMARCA4) in these experiments represent a population average. Therefore, the sharp peaks of CBAF and PBAF, and broader peaks of CBAF and GBAF might actually be representative of distinct nontumor cells within the tumors, and it is not possible to show which have and which do not have SS18::SSX expression. Given the choice of tumor model system, rather than a homogenous cell line system, authors would need to perform single-cell CUT&TAG or another single-cell technique to address this issue.

The principal observations were made on fusion peaks, which are not present in any cells other than those expressing the fusion. These peaks were profiled by ChIP-seq using an antibody specific to the fusion oncoprotein (produced by Kadoch and Hornick [PMID: 32141887] for the clear benefit of the synovial sarcoma community). We have previously reported that the fusion-expressing neoplastic cell population fraction in these mouse tumors is quite high, ranging from 91.1 to 95.9 percent (PMID: 28409421), making the contribution of non-fusion-expressing cells quite low, as well as heterogenous itself (as non-neoplastic cells include macrophages and endothelial cells, among others, which altogether comprise less than 10% of the cells profiled by these bulk epigenomic methods). As these neoplastic percentages are substantial, and one of our senior authors happens to be the chair of the International Human Epigenomes Consortium, guiding all of our bulk epigenomics work to match the IHEC standards, we do not feel that these assessments require single cell CUT&TAG profiling, which is extremely complicated to perform on whole tumor specimens. (We have added a reference to this other work on page 4, Introduction paragraph 4)

3. The results presented in figure 1 – SS18::SSX1 fusion protein localized with H2AK119ub1 in cells – are already well known from previous studies that used patient-derived cancer cell lines and do not add any novel mechanistic insight into the function of SS18::SSX. See PMID: 38177667, PMID: 32747783.

What has not been shown previously, is how this biochemical relationship between SSX and H2AK119ub1 determines the true distribution of the fusion in fusion-driven tumors. We show this here. Although we already cited those papers, we have now added to our Discussion the recognition that the key novelty in this article is our data interrogating that already-reported biochemical relationship on a genomic level *in vivo* in actual tumors, and the way that this interrogation identified another relationship between aberrant canonical BAF distribution and the fusion that has previously received no notice or attention, but will demand additional mechanistic interrogation in the future (page 12, Discussion paragraph 5).

4. Authors probe occupancies of CBAF, PBAF and GBAF specific subunits, DPF2, PBRM1 and BRD9, respectively as a proxy for localization of each complexes. However, it remains unclear as to what

fraction of DPF2 is free subunit and what fraction incorporates into full-assembled complexes in the presence of SS18::SSX fusion protein. DPF2 western blot is missing from the figure 2e. Further, it has been shown previously that DPF2 is not a component of CBAF complexes in the absence of SMARCB1 (see PMID: 29861296, PMID: 28945250, and PMID: 27941797)

Indeed, DPF2 requires SMARCB1 to incorporate into CBAF complexes. We previously demonstrated that nearly all CBAF complexes in synovial sarcoma cells retain both SMARCB1 and DPF2. Please see figures 4-5 and supplemental figures 4, 5 and 7 in the previously published paper (PMID: 34078620). We have also added DPF2 WBs to the prior Fig. 2e (now Fig. 3e) as well as additional demonstration that the same patterns were observed in ARID1A ChIP-seq, in spite of this ChIP-seq showing less robust performance overall, prompting us to use DPF2 as our primary subunit for location of canonical BAF, instead. (Supplementary Fig. 2b-c)

5. Along these lines, what fraction of CBAF and GBAF complexes incorporate SS18::SSX fusion proteins? Is the DPF2 that is pulled-down in ChIP-Seq experiments assembled into CBAF complex or are they 'free' DPF2 simply interacting with nucleosome modifications at active promoters (DPF2 contains double PHD domains)? Clarifying this point is essential to make any claim about the "CBAF complex occupancy" especially since ARID1A/ARID1B ChIP-Seq seemingly did not work in their experiments. Again, homogenous populations of cells are also needed to derive conclusions here.

As above, we previously showed that DPF2 incorporates into CBAF with the fusion and SMARCB1 avidly in synovial sarcoma cell lines (PMID: 34078620). ARID1A ChIP-seq was not as strong as DPF2 ChIP-seq, but we did perform it and demonstrated that the two match each other well. We have also now added analyses to compare these more directly (Supplementary Fig. 2b-c). Most importantly, the use of DPF2 ChIP-seq is a field standard for CBAF complexes, as ARID1A ChIP-seq is challenging in even the best of circumstances (and especially challenging when there is very little of it around in synovial sarcoma cells). As we primarily interpret the DPF2 distribution across chromatin as a loss of the typically distal enhancer binding of DPF2, this result is solid whether or not the binding at the TSS represents free subunit DPF2 or DPF2 incorporated within CBAF. We interpret the DPF2 binding localized to at least some of the promoters as indicating CBAF binding, as these sites have binding of SS18::SSX as well (without any local GBAF binding). While those bound sites could represent both DPF2 and SS18::SSX binding to the sites as individual subunits, the fact that they both are bound to only some of the active TSSs, shows that they are more likely to represent CBAF+fusion binding, at least in that subset.

Further, whether or not the loss of ChIP-seq for DPF2 represents loss of only complete, fully-componented CBAF, or the loss of all CBAF complexes is actually not crucial. It has been shown that partially componented CBAFs that lack SMARCB1 and DPF2 are not fully capable of doing all the work of CBAF complexes (as much of the work of Charlie Roberts in malignant rhabdoid tumors demonstrates). Therefore, even if all that is lost is fully componented CBAF, that still represents a very important loss.

6. Authors should perform ChIP-reChIP experiments with SS18::SSX1 antibody followed by CBAF- and GBAF- specific antibodies to determine the occupancies of these complexes containing SS18::SSX1 fusion protein. Similarly, they would need to perform ChIP-Seq experiments with a SMARCA4 antibody, followed by CBAF/GBAF/PNBAF-specific subunit antibodies to assess the occupancies of CBAF complexes as opposed to free subunits.

While such experiments are simple to suggest, they are extremely difficult to perform even in ChIP performed on cell lines. Further, they would provide little new information beyond the co-occupancies demonstrated by parallel ChIP for each component. As samples for these ChIP experiments are fixed with formaldehyde, whenever a locus is pulled down by one protein, it will drag the other protein along, whether the two are merely co-occupying adjacent positions on chromatin or are complexed together in a protein complex. That same artifact would persist in ChIP-reChIP experiments. Most importantly, we anticipate that they would be prohibitively difficult to perform in tumor-based ChIP-seq.

7. Authors would be expected to validate their findings regarding occupancies of CBAF, GBAF and PBAF complexes in mouse tumors in synovial sarcoma lines. All the experiments in figure 1 and 2 are performed at equilibrium, when the tumor state has already been achieved. Therefore, this model is as much cellularly aplastic as patient-derived cell lines.

Others have performed similar experiments in human cell lines for aspects of this already (PMID: 30431433). Our goal was to expand these observations to the tumor context. We now include fusion-based ChIP-seq in human patient-derived xenograft tumors, which are more representative of real human synovial sarcoma and more directly comparable to mouse synovial sarcomas than long-cultured cell lines. The same phenomenon of broad fusion ChIP-seq peaks correlating with H2AK119ub binding and narrow fusion ChIP-seq peaks lacking these correlations strengthens this phenomenon's presence in another synovial sarcoma context. (Supplementary Fig. 1p-r)

8. Authors claim that DPF2 (and by proxy, CBAF) "avoids H2AK119ub-marked sites" (in abstract), "often accompanying SS18::SSX" (line 205). What is the mechanism by which CBAF containing SS18::SSX, which biochemically interacts with H2AK119ub, avoids H2AK119ub? The finding that SS18::SSX (not wild-type or normal) CBAF would avoid H2AK119ub, while unsubstantiated by any biochemical evidence presented to date, is the only novel potential finding in figures 1 and 2; authors would need to provide a mechanistic basis for this avoidance that is not just a technical artefact of their experiments.

We agree with the reviewer that the mechanism for CBAF containing SS18::SSX to be mostly absent from H2AK119ub-decorated chromatin is a fascinating and much needed pursuit. Biochemical experiments, with which we are also familiar, are not the only data worthy of consideration, when we are trying to understand cancer biology. Indeed, our findings both confirm some biochemical work done by ourselves and others and challenge the extrinsic validity of other biochemical work. We believe that our findings demand additional biochemical investigation, not to prove that they are not artifactual, but to understand the mechanism implied. Such experiments will take years and will fall outside the scope of a mouse genomics and developmental genetics paper.

9. "...CBAF at TSS ****correlates**** with gaps in the H2AK119ub and H3K4me3 distribution..." (line 206) is the only basis for the main conclusion that CBAF avoids H2AK119ub in SyS. However, it appears that CBAF at TSS also correlates with gaps in SMARCC1, which is a subunit in the same complex, at all sites (supplementary Fig 2j at every single example presented). Does this mean CBAF also avoids SMARCC1 in SyS? These data are very weak and correlative and the degree of conclusions derived from these are not substantiated by the data presented.

Perhaps the authors should consider that these conclusions from genomics experiments alone, in absence of any biochemical experiments, are perhaps being over-interpreted.

We have always noted that these experiments are genomic correlations. CBAF does not avoid SMARCC1 in synovial sarcoma. The presence of SMARCC1 appearing to be relatively diminished at those points emphasizes two observations: (1) CBAF complexes comprise an overall lower portion of the BAF family complexes on chromatin in synovial sarcoma, due to the strong recruitment of GBAF complexes bearing the fusion to ubiquitylated H2AK119-decorated chromatin. (2) GBAF that is reported to include two copies of SMARCC1 contrasts with CBAF and PBAF, each of which might include SMARCC2 as the second or even both copies of that subunit. We have removed the SMARCC1 tracks as a distraction in the figures.

10. Authors note that DPF2 (and by proxy CBAF) distributes heavily to promoter regions in SyS, as opposed to murine ES cells. What is the biochemical mechanism by which expression of SS18::SSX leads to such redistribution of DPF2 (or CBAF)? Especially when DPF2 is not even present in CBAF complexes as shown before (see reference above).

There may be an issue with the last sentence. No one has shown that DPF2 is not present in CBAF complexes previously, only that DPF2 requires SMARCB1 incorporation in the CBAF complexes within which it too will incorporate. We have shown previously that DPF2 and SMARCB1 are consistently present in CBAF complexes in synovial sarcoma (PMID: 34078620). As above, we agree that the biochemical mechanism by which fusion expression redistributes CBAF with DPF2 is clearly worthy of pursuit. We feel that this falls outside of the scope of this current paper.

11. Authors claim that CBAF is depleted in SyS (Fig 2e), which was previously reported in another publication from this lab, where knockdown of SSX increased appearance of CBAF complex subunits on glycerol gradient. However, CBAF containing SS18::SSX has been demonstrated to interact strongly with chromatin marked with H2AK119ub even at high salt concentrations in structural and biochemical studies by the Liu and Kadoch labs. Therefore, the apparent loss of CBAF in this study and the other study is very likely an artefact of protein extraction method.

Authors should carefully fractionate chromatin-bound and unbound fractions of nuclear complexes, followed by nuclease treatment to demonstrate if SS18::SSX causes depletion of CBAF complexes. A simple whole cell extract preparation using laemmli buffer might also resolve this issue, as Laemmli buffer efficiently extracts all proteins from chromatin fraction unlike salt.

These issues were thoroughly addressed in our prior paper (PMID: 34078620), to which we refer the Reviewer. Figure 7 in that prior paper clearly demonstrated that while there is an increase of the fraction of CBAF bound to the insoluble chromatin, in the presence of the fusion, this chromatin-bound insoluble fraction is also depleted by the presence of the fusion. Further, whole cell lysates that included nuclease treatments demonstrate the same recovery of CBAF subunits more profoundly than SMARCB1 itself in two cell lines in that figure. Our results do not challenge at all the biochemical interaction between CBAF containing the fusion and H2AK119ub-marked nucleosomes. We celebrate and applaud the elegant work by the Liu and Kadoch laboratories. Our data in that other paper regarding CBAF proteasomal degradation and in this paper demonstrating an altered distribution *in vivo* do not conflict

with the biochemical interaction mechanism; they enrich it with an *in vivo* context. They also demand additional mechanistic work by excellent biochemists in the future to understand how the *in vivo* context changes not the biochemical interactions, but the stability of the CBAF complexes undergoing such interactions.

12. What do the RNA-Seq profiles of human SyS cell lines look like compared to the scRNA-Seq here? Why not do scRNA-Seq in patient derived tumor samples? Isn't that the point of these experiments?

There is value in understanding the process of genetically-induced synovial sarcomagenesis in a living mammalian model. We can profile human tumors and isolate single cells from human tumors as well as cell lines derived from human tumors to learn certain aspects of biology by correlation (which efforts we strongly support, as detailed in our companion paper with human tumor profiling, Hofvander et al., 2024, bioRxiv). However, we will never perform prospective tumorigenesis experiments in the human context. Therefore, we study the process of tumorigenesis using what have serendipitously turned out to be incredibly faithful recapitulations of human synovial sarcomas in the mouse. We have previously compared bulk RNA-seq from our mice to human synovial sarcoma samples, as shown in Figure 2E from our previous paper in *Cancer Discovery* (PMID: 34078620). Others have performed scRNA-seq in human tumors. There are some of these profiles included in our companion paper, as well. We have now added an analysis of our mouse scRNA-seq data compared to published scRNA-seq data from human tumors (PMID: 33495604), showing similar profiles as the new Supplemental Figure 3b.

13. Authors claim that SS18::SSX fusion protein functions predominantly through recruitment of GBAF complex to H2AK119ub, rather than CBAF. However, in rest of manuscript, they delete CBAF- and PBAF-specific subunits but not any GBAF subunit. If their claim is correct, SS18::SSX expression should fail to form SyS in mouse model. Authors would need to generate this mouse model to support their (very strong) claim.

The claim of SS18::SSX distributing with GBAF is an observation from the genomics data from our mouse genetic model, but this has been shown previously in human cell lines (PMID: 30431433), which is why we did not emphasize it as the major initial epigenomics finding that demanded follow-up in this paper. In fact, the principal reason why we present these data is as a validation that these mouse tumors match with an *in vivo* context what has been demonstrated in a human cell line, already, by others. With the co-distribution of SS18::SSX with GBAF an already-accepted phenomenon in the field (PMID: 30431433), we agree that mouse genetic experiments with BRD9 or other GBAF component conditional deletions are warranted, but these were not the focus of this paper, which focused instead on CBAF component losses.

14. Fig 4: Please perform ChIP-Seq for all BAF subunits in *smarcb1*^{f/f} and *wt/wt* cells, not just SMARCA4. Why does SMARCA4 reduce substantially (~2-3x), as loss of SMARCB1 shouldn't have any effect of GBAF and CBAF, which are enriched at these sites?

While the proposed experiments would be potentially interesting, they are not pertinent to synovial sarcoma, as the combination of *Smarcb1*-silencing and SS18::SSX expression does not generate synovial sarcomas, as has been previously shown (PMID: 34078620) and is again shown here. We are not entirely sure what Reviewer #1 means by suggesting that SMARCB1 loss would have no effect on CBAF, in the last sentence above. SMARCB1 is a standard component of CBAF. CBAF has been shown to be lost

dramatically at proximal sites particularly in the absence of SMARCB1 (PMID: 27941797), to which mechanism its added genetic silencing in the setting of fusion expression probably hearkens. While CBAF levels overall are somewhat recovered with the combination of *Smarcb1* silencing and fusion expression, these CBAFs are not fully-componented or fully-functional. Our clear observation is that these SMARCB1-less CBAFs no longer distribute to these proximal sites, similar to the lost distribution of SMARCB1-less PBAFs. The phenomenon that we observe here is that GBAF with the fusion does not distribute to the TSS specifically, which is why SMARCB1 loss renders an overall gap at the TSS through alteration of PBAF and CBAF distributions there. To delve into this phenomenon more deeply may have pertinence to GBAF general biology or possibly to H2AK119ub biology (which also is missing from the active TSS, typically), but we feel it departs from the major point of this manuscript. It also strengthens the idea that DPF2 bound at these TSSs is not only free subunit DPF2, but DPF2 incorporated into CBAF complexes in the SMARCB1-intact synovial sarcoma context (pertinent to point #5, above).

15. PBAF complexes localize to the promoters of active genes and sustain their transcription in all cells, including SyS. Unsurprisingly, PBRM1 knockout in their SyS model altered the typical SyS phenotype. What is the relevance of these findings in SyS?

We fully agree with the reviewer on the normal function of PBAF. The interesting observation from our prior paper is that PBAF abundance is increased in the presence of the fusion. The principal importance of these genetic experiments with PBRM1 loss is as a comparison to ARID1A or ARID1B loss. This comparison is important as PBRM1, like SMARCB1, ARID1A and ARID1B, is considered to be a frequently silenced tumor suppressor gene among the genes coding for BAF family subunits. Also importantly, human synovial sarcomas with secondary mutations in *PBRM1* and *SMARCB1* have been reported, as have cases with loss of *ARID1A* or *ARID1B*. We have now added an analysis of human mutational profiles in reported synovial sarcomas for these genes (Supplementary Fig. 10b).

16. What is the mechanism by which ARID1A or ARID1B disruptions enhance SyS tumorigenesis {Note that CBAF is not actually disrupted in either of these cases because of paralog redundancy; see below}? ARID1A or ARID1B loss enhances SyS sarcomagenesis. However, LOF mutations in these proteins are never found in early- or late- stage SyS tumors from human patient, which would be a prediction from this study?

Very few secondary mutations have been identified consistently in human synovial sarcomas, which have a very low mutational burden overall, even compared to other fusion-driven sarcomas. We do not predict from our genetic experiment in mice that such mutations would necessarily be identified in human synovial sarcomas. Whether or not CBAF is fully disrupted—which we agree, it is not—the loss of ARID1A function is a powerfully oncogenic single-gene event in many types of cancer, and represents an important loss of function for CBAF complexes, even if incomplete. The fact that this slightly enhances tumor formation without substantially changing the synovial sarcoma transcriptome or phenotype suggests that ARID1A functional depletion is a protein-level consequence of the fusion expression alone. Also, pertinent to an answer to Reviewer #3's comment below, the observation of *ARID1A* and *ARID1B* mutations in human synovial sarcomas, in addition to mutations in *PBRM1* and *SMARCB1* have been reported. We have added the pertinent data from cBioPortal to the revised manuscript (Supplementary Fig. 10b) and thank the Reviewer for highlighting this oversight in our previous submission.

17. ARID1A and ARID1B are two paralogs, that both each assemble into CBAF complexes mutually exclusively. Likewise, loss of one can be compensated by the other paralog as has been shown (Helming et al., Nat Med 2013). Authors only deleted one of the two paralogs in all the experiments performed in figure 6, never both. Therefore, CBAF is not necessarily disrupted in these cells as the remaining paralog compensates for the loss of one. In the absence of ARID1A/ARID1B double knockout, the results presented in figure 6 and 7 are un-interpretable.

While ARID1A and ARID1B are paralogs for their inclusion in CBAF complexes, they are not perfectly paralogous in their biological function, where ARID1A genetic loss is the most common BAF-related genetic finding across all cancers. The paper cited (PMID: 24562383) is a tour de force review of the mutational spectrum across many cancers, demonstrating that the loss of both copies of *ARID1A* and both copies of *ARID1B* is extremely rare in human cancers. This indicates that almost never is the complete loss of BAF family complexes tolerated by mammalian cells. It confirms that these two alleles may mutually compensate at some very basic level for each other. It does not refute or even challenge the idea that loss of both copies of either *ARID1A* or *ARID1B* has a profound effect on most cells. Our genetic experiments unexpectedly demonstrated that loss of either does not have a clearly discernible effect on synovial sarcomagenesis, other than slightly accelerating it. (pages 12-13, Discussion paragraphs 4-6)

We disagree that these results are “uninterpretable,” but have adjusted what interpretations are possible, given the clear findings. We have adjusted the language of our interpretation of these results to focus on ARID1A and ARID1B directly, as CBAF members, without claiming entire CBAF disruption, which is clearly not present in these genetic tumorigenesis experiments (pages 12-13, Discussion paragraphs 4-6).

18. Authors should test these genetic findings presented in Fig 4-7 in patient-derived SyS cell lines. Does loss of ARID1A and ARID1B enhance proliferation of SyS cell lines? Does PBRM1 deletion in SyS cell lines change their phenotype at all?

Genetic tumorigenesis experiments in mice are not directly predicted to be reflected by changes in proliferation in human cell line disruptions, which is why we did the experiments in mice. What has been shown in the DepMap is that disruption of *ARID1A*, *ARID1B*, and *PBRM1* had minimal effects on synovial sarcoma cell line proliferation. The key to our doing these experiments in mice developing tumors is that we are asking whether loss of these genes impacts the development of tumors, not what it does to fully formed tumor cells. In fact, we would predict from our data that the loss of *ARID1A* or *ARID1B* would not change proliferation of synovial sarcoma cell lines, significantly, as the general dysfunction of CBAF is already achieved at the protein level in fully developed synovial sarcomas. We have added the pertinent DepMap data from human cell lines to Supplemental Fig. 10c. Please also see pages 12-13, Discussion paragraphs 4-6.

Reviewer #2 (Remarks to the Author): Expert in sarcoma, gene fusion, epigenetics, BAF complex and mouse models

Li and colleagues addressed the complex issue of epigenomic dysregulation mechanisms in synovial sarcoma (SyS).

The SS18::SSX oncoprotein substitutes SS18 in CBAF and GBAF complexes, leading to preferential CBAF reduction as previously shown by the same research group (Li et al., Cancer Discov, 2021) and genome wide altered location of GBAF complexes. The SSX domain of the fusion plays a role in GBAF redistribution, indeed the SSX domain directly binds in a peculiar way to nucleosomes which are marked with H2AK119Ub1 (Tong et al., Nat Struct Mol Biol, 2024) for silencing.

In this paper, the Authors show that SS18::SSX targets promoters in two patterns, narrow and broad, with broad promoter binding for sarcomagenesis activated transcription (SAT) genes and narrow binding for genes that are maintained active before and after transformation (MAT genes). SS18::SSX expression redistributes GBAF complexes broadly to promoters and distal enhancers marked by H2AK119ub as evinced by the strong coincidence of H2AK119ub and the fusion in histone mark and fusion CHIP-seq enrichment profiles. Binding of GBAF complex causes H3K27me3 loss and makes transcriptionally active those developmental loci, previously marked for silencing, leading to inappropriate activation of developmental genes that induce SyS onset and progression. In addition CBAF containing SS18::SSX avoids H2AK119ub-marked sites, and instead distributes with PBAF narrowly to transcription start sites as highlighted by the strong correlations found between PBAF and CBAF distributions in these sites.

The Authors frequently quote and refer to two paired papers from their own group not yet published but only available online as preprints on bioRxiv. In one of these papers, namely in "Synovial Sarcoma Chromatin Dynamics Reveal a Continuum in SS18::SSX Reprogramming" by Hofvander et al, they analyze the mechanisms of action of the fusion transcript SS18::SSX in human synovial sarcoma. The present work further analyzes transcriptional reprogramming by exploiting synovial sarcomagenesis in mouse models. The Authors take advantage of different SyS conditional mouse models on hSS2 (SS18::SSX2) or hSS1 (SS18::SSX1) background with genetic disruption of Smarcb1 (PBAF and CBAF component), or Pbrm1 (PBAF component), or Arid1a and Arid1b (both CBAF components), to investigate how each of them impacts on CBAF, GBAF and PBAF destiny and activity, and affects tumorigenesis, adding insights into the role of different components of BAF chromatin remodeling complexes. Loss of these elements speeded up tumorigenesis in hSS2 mice, loss of Smarcb1 and Pbrm1 changed tumor morphology while loss of Arid1a and b did not. They conclude that synovial sarcomagenesis requires both SS18::SSX action on transcription reprogramming through GBAF-mediated reactivation to polycomb-targeted developmental genes, and reduced presence and altered localization of CBAF. They further suggest that the focal localization of CBAF and PBAF at the TSS requires SMARCB1.

Overall results are convincing, significant and relevant to the topic. The work is original and further develops established literature. The methodology is largely correct and meets the accepted standards in the field. Details are fully provided in the methods for the work to be replicated.

However, the following points are identified and need to be addressed for revision before publication in Nat Commun.

Major points:

- 1) The Authors cite to two paired papers from their own group not yet published but only available online as preprints on bioRxiv. Indeed, the reading of these two articles is of help to the understanding of the present paper and they should be more extensively recapitulated in the paper. The authors should better describe how and in which model the three gene categories (SAT, MET and SST) were identified.

We have added more extensive explanation of the key findings from our companion papers and the definition of the three gene categories. This is actually quite critical to the understanding of the paper and is also noted above in the response to a Reviewer #1 critique stating that we had no control cells. These critical experiments in our companion paper (Hill *et al.*, bioRxiv, 2024; *Nature Communications*, accepted) identified the best control cells we have ever had for synovial sarcomagenesis, and are now presented in summary fashion in Figure 1, to introduce the concepts of the three gene categories. The directly pertinent results of the other paper are also summarized in this Figure.

2) The hSS1;Pbrm1-fl/fl mice behave differently from hSS2;Pbrm1-fl/fl mice as regard tumorigenesis. The Authors conclude that “this apparent paradox suggests that Pbrm1 silencing similarly blunted synovial sarcomagenesis programs in both SS18::SSX backgrounds. This blunting of SyS programs tipped the faster growing hSS2 tumors into an alternate tumorigenesis program, but merely slowed hSS1 tumor development, causing selection for alleles that escaped recombination”. Any role of a differential immune signature between the two backgrounds?

We have looked into the transcriptomes of these two tumor groups more carefully and identified few important changes in transcription between them, probably because we are examining only fully formed tumors by the time these samples are obtained (Supplementary Fig. 7h).

3) The Authors state that: “Having firmly established the expression of SAT gene signatures in human SyS in our related manuscripts^{29, 31}, we next sought to confirm that these CBAF-loss related signatures of gene expression and repression are also conserved between species with SS18::SSX-driven malignancy. The SST genes that were silenced more profoundly by Arid1a genetic disruption were found to be very lowly expressed in 52 human SyS transcriptomes. Inversely, the pseudo-time trajectory associated MAT pattern genes that were further upregulated by Arid1a silencing were highly expressed in the same human SyS tumors”. Are these “MAT pattern genes” involved in SyS tumor progression?

MAT pattern genes generally are genes that have a typical housekeeping gene pattern of histone marks, suggesting that they are expressed in both the cells of origin and the fully transformed sarcoma cells. The particular subset of MAT pattern genes discussed in the prior Fig. 7 (now Fig. 8) is apparently involved with synovial sarcomagenesis, as they are further upregulated in the pseudo-time trajectory between early sarcoma cells and those that have reached a more fully reprogrammed state of poorly differentiated status. Because we propose that these genes are indirectly activated by the expression of the fusion oncoprotein (in that they lack the chromatin patterns indicative of SSX binding to ubiquitin in broad regions) they are harder to identify discretely across species and across different models in the same species. Nonetheless, we provide the data from human tumors as supportive evidence that these genes—indirectly targeted by disruptions in CBAF—are a critical part of the synovial sarcoma transcriptome (See Fig. 8d and the newly added Supplementary Fig. 10a). We believe that this is the most important finding of our work in this paper. Most of the biochemistry, genomics, and even BAF directed drug-discovery work in the field of synovial sarcoma has focused on the genes whose expression is upregulated by the fusion oncoprotein binding at their promoters. However, the data we present here have uncovered another pattern of transcriptome alteration that also appears to be conserved across multiple synovial sarcoma contexts and important. Certainly, future work will test the relative contributions of each of these mechanisms of gene regulation driven directly or indirectly by the

fusion oncoprotein, as well as the biochemical mechanisms behind this newly fleshed out CBAF dysregulation.

3) “As future therapeutic efforts variably target each of these mechanisms (gain of GBAF function at polycomb targets or loss of CBAF function), the gene sets we have identified as associated with each constitute vital datasets to interrogate mechanisms and efficacies of therapeutics”. This issue should be expanded mentioning possible drugs targeting H3K4me3, or H2AK119ub, or BAF complexes, and eventually related gene expression data if available.

There remains much to be worked out with regard to therapeutics in synovial sarcoma. However, there has already been development of PROTAC degraders of BRD9, attempting to target the GBAF gain of function and there is a report in press at *Nature Communications* and already available on *bioRxiv* that demonstrates the therapeutic potential of CBAF restoration. Another paper published recently in *Science Advances* indicates the impact of a WDR5 PROTAC on synovial sarcoma cell lines and xenografts (PMID: 40267190. WDR5 catalyzes the placement of H3K4me3 marks). We have added references to each of these therapeutic strategies to the Discussion (page 13, Discussion paragraph 8).

Minor points:

1) In Figure 1, b,c,d,e, drawings need to be arranged in a row or grouped in order to be more readable

We have rearranged this figure to group these four panels together for more direct comparison.

2) In the results, with reference to Figure 1, h-m, please comment on H3K36me3

We have added discussion of H3K36me3 (page 4, Results paragraph 5), which directly marks actively transcribed chromatin in gene bodies, as opposed to promoters or enhancers, and therefore serves as a control for other histone post-translational modifications in this paper.

3) In Figure 3, a, please check for the indications of the five source tissue types

The tissue types of the source tissues for the scRNA-seq experiments were determined by frozen section pathology performed on the adjacent tissues at the time of harvest. This is now added as Supplemental Fig. 3a.

4) In Figure 6, f, please check the nomenclature of each picture

We have now labeled each of these photomicrographs, which are now Fig. 7f.

5) In methods “Single cell RNA-seq” : “Cells were filtered based on quality metrics, including removal of cells with mitochondrial features surpassing 25%, or total features below 750.” Please explain why using the parameter of 25% and not the more selective 10% or 5%?

We have added the following explanation to the Methods:

“We plotted the distribution of mitochondrial percentages and nFeature_RNA values to guide threshold selection, and then tested different cutoffs to remove poor-quality and artifactual events. Very stringent cutoffs (5–10%) tended to exclude an excessive number of cells, including metabolically active populations such as malignant and stromal cells. Key results remained stable across thresholds of 20–30%. Because our samples were derived from fresh solid tumors, enzymatic dissociation likely led to elevated mitochondrial transcript fractions by preferentially depleting cytosolic RNA. We therefore applied a relaxed global filter of $\%mt \leq 25\%$, which removed overtly damaged cells while retaining true tumor and microenvironmental populations. To further avoid including low-quality cells, cells with fewer than 750 detected genes were removed, and DoubletFinder was applied to identify and discard putative doublets.”

6) Please make sure that References are reported uniformly and according to the journal requirements.

We have thoroughly checked our references for completeness and accuracy. Two of the cited papers are accepted and pending publication by *Nature Communications*, which will permit the updating of these references as soon as publications are complete (Hill, et al. and Floros, et al.)

Reviewer #3 (Remarks to the Author): Expert in sarcoma, mouse models, epigenetics and ChIP-seq

The manuscript by Li and Li et al. entitled “Synovial sarcoma reprograms transcription by GBAF activation of polycomb targets and loss of CBAF enhancers” uses a previously described transgenic mouse model for the SS18::SSX oncogene – the dominant driver of synovial sarcoma (SyS) to study how this fusion oncoprotein promotes a marked redistribution of non-canonical GBAF complexes to promoters and distal enhancers marked by H2AK119ub, which causes so-called “developmental loci” to lose H3K27me3 and consequently become active. Using a series of ChIP-seq and HiChIP experiments, the authors demonstrate that canonical BAF complexes containing SS18::SSX abandons its typical binding sites and redistributes close to PBAF at TSSs. For this phenotype, the CBAF-specific components Arid1a or Arid1b appear to be not relevant as the SyS phenotype in mice is largely unaltered. However, the PBAF- and CBAF-specific components Smarcb1 or Pbrm1 (PBAF-specific) has an impact on SyS formation with accelerated tumorigenesis driven by SS18::SSX. In sum, this is an elegant study combining state-of-the-art omics technologies to uncover how SS18::SSX reprograms transcription through GBAF redistribution in transgenic SyS mouse model. The paper is well-written, and the conclusions are largely supported by the data. There are only a few concerns/aspects that should be addressed before publication:

Major comments:

1. The single-cell and spatial transcriptomic data shown in Figure 3 are very interesting. Yet, it would be advisable to confirm the spatial expression of some of the proposed markers at the protein level by IHC in tumor tissue.

We have added immune profiling for the proteins expressed from two of the spatially transcribed genes for some of these markers of tumor progression as Supplementary Fig. 3c.

2. While this study certainly has the groundwork for more translational studies, the current manuscript could benefit from some analyses to which the authors already allude to in lines 390-392: “This observation fits a model for heterogeneity in human SyS, where tumors that have more genomic copy number variation also have more retained bivalency, as shown in our human datasets.” Do SyS tumors also show in some cases alterations of SMARCB1 and/or PBRM1 that could translate into variable clinical phenotypes?

Some of this needs to be deferred to our companion paper (Hofvander et al, 2024, bioRxiv), which delves into these issues deeply and directly. We have added the findings of SMARCB1 and PBRM1 mutations that have been identified in human synovial sarcomas (Supplementary Fig. 10b). Nature has supplied too few of these variants to study them as subset populations, but their existence is telling, in that SMARCB1 and PBRM1 losses do “accomplish something different” in the synovial sarcoma context than otherwise is native to the biology of the fusion itself.

Can the authors elaborate on which CNVs may contribute specifically to the more retained bivalency?

Again, this is more deeply explored in the companion paper, but our working model is that bivalency is retained or regained at more loci when full reprogramming by the fusion is less critical to the transformed state of the cancer cells.

Are the transcriptional features and signatures shown in Fig. 7 of prognostic relevance?

We have added an analysis of the prognostic impact of the different expression signatures in the human synovial sarcoma dataset reported in our other paper as Supplementary Fig. 10a.

3. Recent data in epithelioid sarcoma – a SMARCB1 deficient cancer – has shown that the residual SWI/SNF complex has an altered but still active effect on the transcriptional and epigenetic reprogramming of these tumor cells that may constitute a specific vulnerability of BRG1 inhibition (PMID 39834137). It would be very interesting to explore whether SS18::SSX confers a similar vulnerability to SyS cells/tumors.

We fully agree that BRG1 inhibition as a vulnerability is an interesting biology to investigate in synovial sarcoma. BAF enzymatic activity also represents an oddity in the biology of synovial sarcoma, that is not yet fully explained. In the DepMap, it is demonstrated that BRG1 (SMARCA4) depletion is minimally impactful on synovial sarcoma cell lines (Supplementary Fig. 10c). This may relate to the presence of an enzymatic paralog in BRM. However, both of these have been degraded by non-specific PROTACs by other investigators and found to have minimal impact on some of the tested synovial sarcoma cell lines. We have seen these data presented in closed conferences. As these are not our data to share, we will eagerly await—with Reviewer #3—the publication of these results by other investigators. We have also pursued genetic silencing of BRG1 or BRM in our synovial sarcoma mice, which produces interesting, but complicated, results that will require more space to fully develop in a separate manuscript.

Minor comments

1. Line 32 abstract: should read SS18::SSX (not SS18:SSX)

This has been corrected.

2. Line 50: „...expression of one of these fusions...” >> Which one(s)?

This has been explained in the text.